# Omni-MATH: A Universal Olympiad Level Mathematic Benchmark for Large Language Models

**Bofei Gao**[1*]**, Feifan Song**[1]**, Zhe Yang**[1]**, Zefan Cai**[2]**, Yibo Miao**[4]**, Qingxiu Dong**[1]**, Lei Li**[9]**,
**Chenghao Ma**[5]**, Liang Chen**[1]**, Runxin Xu**[1]**, Zhengyang Tang**[6]**, Benyou Wang**[6]**, Daoguang Zan**[7]**,
**Shanghaoran Quan**[3]**, Ge Zhang**[8]**, Lei Sha**[10]**,Yichang Zhang**[3]**, Xuancheng Ren**[3]**,
**Tianyu Liu**[3*]**,Baobao Chang**[1†]

[1]Peking University    [2]University of Wisconsin - Madison    [3]Alibaba Group
[4]Shanghai Jiao Tong University    [5]Engineering Research Center of Information Networks
[6]The Chinese University of Hong Kong, Shenzhen
[7]Institute of Software, Chinese Academy of Sciences
[8]University of Waterloo
[9]The University of Hong Kong
[10]Zhongguancun Laboratory

## Abstract

Recent advancements in large language models (LLMs) have led to significant breakthroughs in mathematical reasoning capabilities. However, existing benchmarks like GSM8K or MATH are now being solved with high accuracy (e.g., OpenAI o1 achieves 94.8% on MATH dataset), indicating their inadequacy for truly challenging these models. To bridge this gap, we propose a comprehensive and challenging benchmark specifically designed to assess LLMs' mathematical reasoning at the Olympiad level. Unlike existing Olympiad-related benchmarks, our dataset focuses exclusively on mathematics and comprises a vast collection of 4428 competition-level problems with rigorous human annotation. These problems are meticulously categorized into over 33 sub-domains and span more than 10 distinct difficulty levels, enabling a holistic assessment of model performance in Olympiad-mathematical reasoning. Furthermore, we conducted an in-depth analysis based on this benchmark. Our experimental results show that even the most advanced models, OpenAI o1-mini and OpenAI o1-preview, struggle with highly challenging Olympiad-level problems, with 60.54% and 52.55% accuracy, highlighting significant challenges in Olympiad-level mathematical reasoning.

| | | |
|---|---|---|
| **Github Repo** | [GitHub Page] |
| **Rule-based Repo** | [GitHub Page] |
| **Project Page** | [Project Page & Leaderboard] |
| **Dataset** | [Huggingface Dataset] |
| **OmniJudge** | [Huggingface Model] |

## 1 Introduction

Large Language Models (LLMs) (OpenAI, 2023; Abhimanyu Dubey & Abhishek Kadian, 2024; Yang et al., 2024a) have shown remarkable capabilities in producing human-like text (Abhimanyu Dubey & Abhishek Kadian, 2024), code generation (Hui et al., 2024; Rozière et al., 2024), and dialogue (Chiang et al., 2024). Among these capabilities, mathematical ability serves as a fundamental measure of the problem-solving and complex reasoning skills of LLMs, motivating various endeavors towards improving the mathematical reasoning of LLMs (Yang et al., 2024b; Azerbayev et al., 2024; Wang et al., 2024).

---

*Project Lead.

†Corresponding Author.

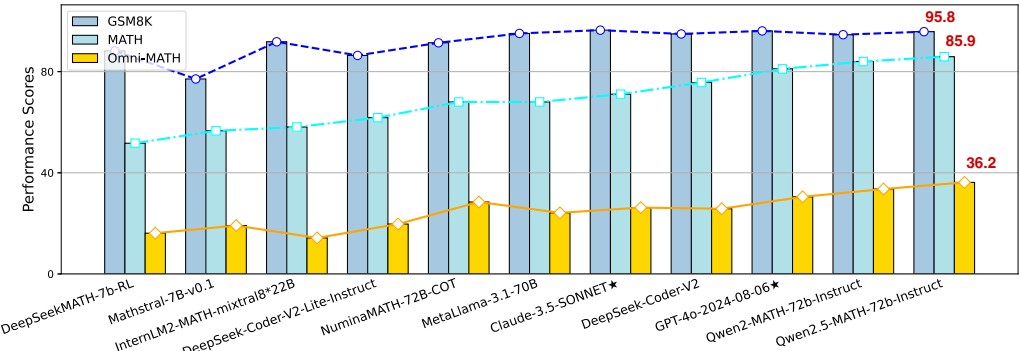

Figure 1: Comparisons among different models on GSM8K, MATH, and **Omni-MATH**, where the models are ranked based on their performance on MATH, and those marked with "⋆" are closed-source models. As observed, the iterative advancements of these models show that existing benchmarks are nearing saturation. Our proposed Omni-MATH introduces a challenging benchmark to further advance mathematical intelligence in large language models.

To evaluate the mathematical capabilities of LLMs, researchers have introduced various mathematical benchmarks (Zhang et al., 2024), including the well-recognized GSM8k (Cobbe et al., 2021) and MATH (Hendrycks et al., 2021). However, as LLMs have evolved rapidly, these benchmarks have increasingly lost their challenge (OpenAI, 2024), resulting in a diminished capacity to differentiate between model capabilities, as illustrated in Figure 1. Although some efforts have been made to establish more challenging benchmarks, most prior work has not focused specifically on evaluating mathematical capabilities (He et al., 2024). In addition, the test results are closely coupled with other LLM capabilities, such as multimodal fusion, rendering the findings to be vague for understanding the mathematical reasoning improvements (Huang et al., 2024).

To bridge this gap, we introduce Omni-MATH, a universal Olympiad-level benchmark specifically designed for mathematical reasoning. Our dataset exclusively focuses on text-only mathematics and encompasses an extensive collection of 4,428 competition-level problems. We curate our benchmark based on the actual selection processes of mathematical Olympiads, ensuring that our dataset includes problems from both introductory competitions and professional international contests. To further explore LLMs' performance in multiple mathematical areas, we have established a hierarchical classification of mathematical domains, including 33 sub-domains and more than 10 distinct difficulty levels (see Figure 2), allowing for a nuanced analysis of model performance across various mathematical disciplines and levels of complexity. For better evaluation, we provide a GPT-4o-based model evaluation and an open-source verifier, Omni-Judge, besides the traditional rule-based evaluation. Our Omni-Judge provides achieves over 91% consistency with GPT-4o and 86% consistency with human judgments, providing reliable feedback. What's more, the open-source verifier provides a simple and effective way to evaluate the Olympiad-level mathematical problems.

We conduct a comprehensive evaluation of the currently strongest models on Omni-MATH. As shown in Table 3, it presents a significant challenge to the existing models. Even the most capable models OpenAI o1-mini and o1-preview, renowned for their reasoning ability, only achieve an accuracy of 60.54% and 52.55%, while the highest score among open-source models is just 36.2%. In addition, the detailed categorization of mathematical disciplines and comprehensive evaluation provide new insights into current LLMs. For instance, LLMs show a marginally greater aptitude for solving algebra, while struggling significantly with discrete mathematics. The commonly used approach for test-time scaling, Best-of-N, has proven ineffective for Olympiad-level mathematics problems. This underscores the urgent need for further research into test-time scaling techniques. To summarize, our contribution includes:

1) We introduce Omni-Math, a universal Olympiad-level mathematical benchmark with over 33 sub-domains and diverse difficulty levels, posing new challenges to the problem solving and complex reasoning capability of LLMs.

2) We comprehensively examine different evaluation models. Our experiments reveal that GPT-4o evaluation can align well with human evaluation with an accuracy of 98% and the Omni-Judge

achieves over 90% consistency with GPT-4o, providing an efficient and reliable evaluation.
3) We conducted a comprehensive evaluation of 15 LLMs and found that even the most advanced models, OpenAI o1, struggle with highly challenging problems, with 60.54% accuracy. Additionally, LLMs frequently fail to solve discrete mathematics problems and often make logical missteps, highlighting significant challenges in mathematical reasoning and problem-solving for LLMs.

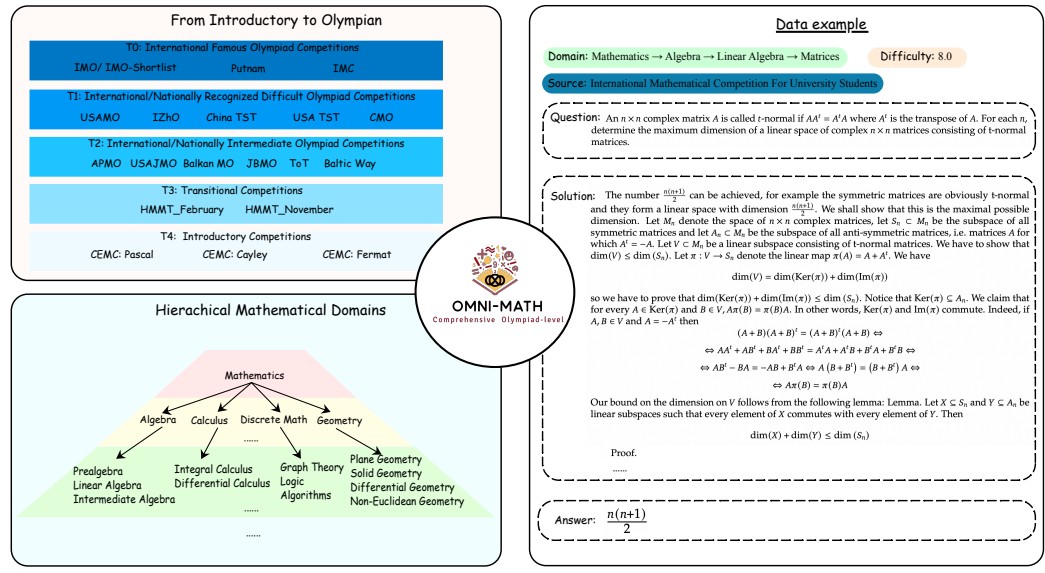

Figure 2: An overall illustration of Omni-MATH. The left-top part presents the diverse and well-structured data sources of Omni-MATH. The left-bottom part represents the hierarchical mathematical domains of Omni-Math. The right part presents a concrete data example of Omni-Math.

Table 1: Comparison of various mathematical reasoning benchmarks. "# Data" denotes the total number of data in the corresponding benchmark, "# T.M" denotes the number of textual English mathematical reasoning data, "# T.O.M" indicates the number of textual English Olympiad-level mathematical reasoning data, "# Domain" refers to the number of domains in Olympiad-level mathematical data, "# Diff" represents the number of difficulty levels, "Leak Det." signifies whether data leakage detection is performed and "Evaluator" describes the evaluation methods employed.

| Name | # Data | # T.M. | # T.O.M | # Domain | # Diff. | Leak Det. | Evaluation |
|---|---|---|---|---|---|---|---|
| GSM8K | 1319 | 1319 | 0 | - | - | ✗ | Rule |
| MATH | 5000 | **5000** | 0 | 6 | 5 | ✗ | Rule |
| JEEBench | 515 | 515 | 0 | 5 | - | ✗ | Rule |
| CHAMP | 270 | 270 | 270 | 5 | - | ✗ | Rule |
| Olympiad Bench | 8476 | 675 | 675 | 3 | - | ✗ | Rule |
| MATH Odyssey | 387 | 387 | 148 | 12 | 4 | ✗ | GPT-4 |
| Olympic Arena | **11163** | 2036 | 2036 | - | - | ✓ | Rule & GPT-4 |
| Omni-MATH | 4428 | 4428 | **4428** | **33+** | **10+** | ✓ | OmniJudge (Ours) & GPT-4o |

## 2 OMNI-MATH BENCHMARK

In this section, we explore the construction process of Omni-MATH. This involves data generation, manual annotation(§E.1), and the classification of domains (§2.3) and difficulty levels (§2.2). Furthermore, we will introduce the evaluation (§2.4) of the Omni-MATH, which includes GPT-4o-based evaluation and the open-source evaluator Omni-Judge.

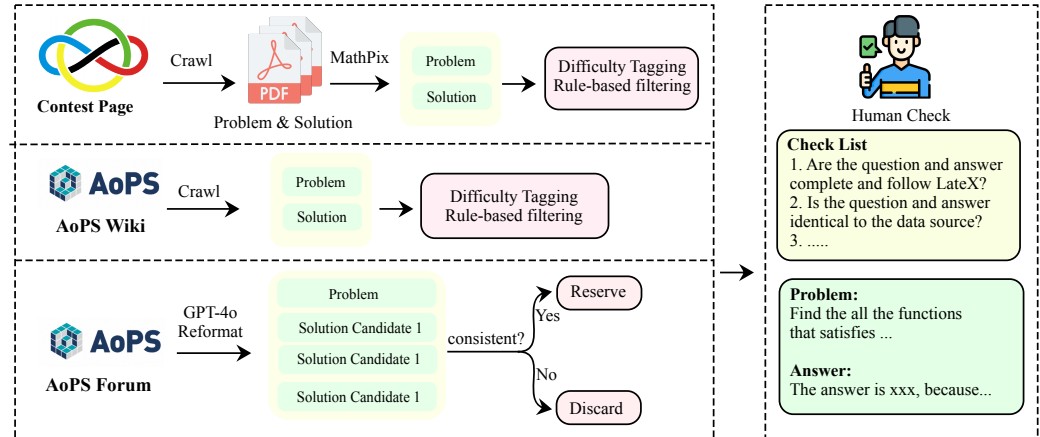

Figure 3: The overall data collection and annotation process of Omni-MATH.

## 2.1 DATA COLLECTION AND ANNOTATION

First, we conducted a comprehensive review of mathematics competitions worldwide and classified them into five levels based on difficulty, scale, and prestige, referencing the website *AoPS Wiki: Competition Ratings* and other forums containing the discussions of mathematical contests. The details are presented in Table 5. It is important to note that this classification is somewhat preliminary; it serves only to determine which competitions we include in our benchmark, rather than providing definitive judgments on the difficulty level of each problem. After obtaining these target contests, we start our data collection and annotation process, as shown in Figure 3.

**Data Collection**  For some target competitions like IMO and IMC, we can directly crawl the publicly available problems and solutions from the official website. Then we utilized MathPix to convert the PDF documents of the problems and solutions into LaTeX format. Given that the provided solutions are guaranteed to be correct, we prioritized crawling these solutions for data collection.

For sections lacking open-source solutions, we extracted data from the famous AoPS website, which includes information from both the *AoPS Wiki*[1] and the *AoPS forum*[2]. The *AoPS Wiki* features recognized solutions, whereas solutions in the *AoPS forum* are often user-uploaded and may not be correct. We crawl the problem with multiple user-uploaded solutions following methods detailed in Appendix D. A rough consistency check is conducted by extracting final answers and checking whether they are identical to each other, leading to the exclusion of any inconsistent cases. Additionally, cases from which fewer than three answers could be scraped were also discarded. For each data source, we applied rules for initial filtering after obtaining the data, ensuring that both the questions and responses generally adhered to the specified input and output formats.

**Data Annotation**  After the automated construction of all the data, we engaged a team of professional annotators, comprised of graduate and doctoral students to verify the solutions and the answers of the dataset manually.

For the data from *Contest Page* and *AoPS Wiki* sections, which included official solutions, the verification process is relatively simple. We focus on ensuring consistency with the respective data sources. Conversely, we place particular emphasis on the problems sourced from the *AoPS Forum*. Following an initial screening,

Table 2: The number of data in the construction process. "#Data" denotes the number of data, "#R.Filter" denotes the number of data after rule-based filtering, and "#H.Filter" denotes the number of data after human-annotation.

| Data Source | #Data | #R.Filter | #H.Filter |
|---|---|---|---|
| Contest Page | 3750 | 3596 | 3242 |
| AoPS Wiki | 8575 | 298 | 298 |
| AoPS Forum | 17502 | 1100 | 888 |

[1]https://artofproblemsolving.com/wiki/index.php
[2]https://artofproblemsolving.com/community/c13_contests

we narrow the dataset down to 1,100 problems.

At this stage, we engage a team of four annotators to assess whether the most frequent responses from the original forum align with the final outputs produced. Each annotator reviews 550 problems, ensuring that each problem is analyzed by two different personnel for greater reliability. We employ cross-validation to enhance the robustness of our findings, yielding an accuracy rate of 92.7%. After excluding inconsistent cases, we conducted a manual sampling of 150 entries, resulting in an improved accuracy rate of 97.3%. The overall data generation process is shown in Table 2, and additional annotation specifics can be found in Appendix C.

After the data collection and data annotation section, we obtain the overall data for Omni-MATH.

## 2.2 DIFFICULTY CLASSIFICATION

Through practical research, we find out that the difficulty of Olympic-level problems varies significantly. To enable a comprehensive assessment of Olympiad-level mathematical reasoning across various levels, We categorize the overall data by different difficulty levels. Relying solely on factors such as the difficulty and popularity of competitions to determine problem difficulty is insufficient. To achieve a more objective assessment of the difficulty of each problem in our dataset, we consulted the *AoPS: Rating* [3] page, which provides difficulty ratings for problems from various competitions. Specifically, the difficulty is quantified on a scale from 0 to 10, including increments such as 0.5 and 0.25. For competitions listed on this page, we directly assign the corresponding difficulty score on the problem. For those competitions not included, we utilize the existing competitions, problems, and their associated difficulties on this page as a basis for in-context learning and prompt GPT-4o to assign specific difficulty ratings. The details of this prompting procedure can be found in the Appendix E.4. Following categorization, the overall distribution of difficulty levels is presented in Figure 4.

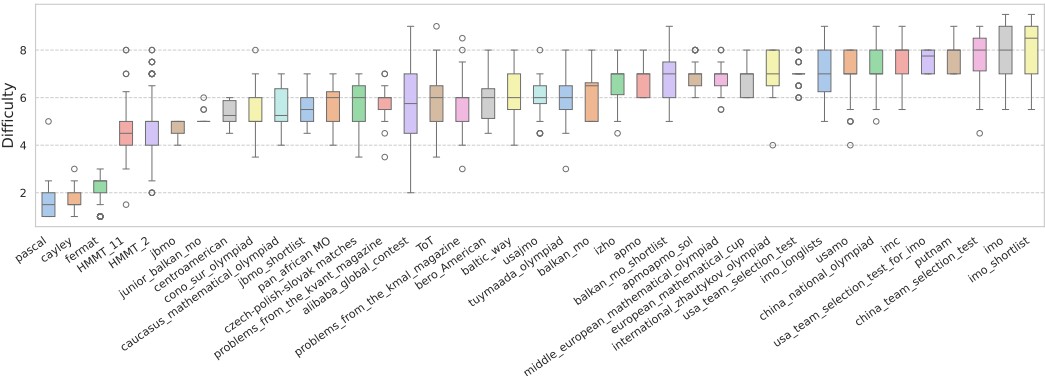

Figure 4: Difficulty distribution across contests.

After conducting instance-level difficulty classification, we organized the difficulty distribution by competition. Our analysis revealed that the actual difficulty distribution rankings shown in Figure 4 closely align with the tiered classifications we previously surveyed in Table 5. This consistency further validates the reliability of our domain classification.

## 2.3 DOMAIN CLASSIFICATION

Unlike existing benchmarks, we argue that Olympic-level mathematics competitions encompass a wide range of complex topics. The data from different domains present a greater challenge for model generalization, and it is quite likely that a single training set will yield inconsistent improvements across various fields. Therefore, it is essential to categorize the existing dataset into distinct mathematical domains to better investigate the model's performance across different math areas.

---

[3]https://artofproblemsolving.com/wiki/index.php/AoPS_Wiki:Competition_ratings

Specifically, we draw upon relevant competition guidebooks (Engel, 1998; Zeitz, 2017) to organize the mathematical fields into a hierarchical tree structure. This tree-based structure not only enhances the rationality of domain classification by aligning with the table of contents of reference materials but also allows us to compute accuracy metrics at varying levels of granularity.

For the domain categorization, we utilized GPT-4o to classify the problems. The prompt used for classification can be found in Appendix E.5, along with the detailed classification results.

## 2.4 EVALUATION

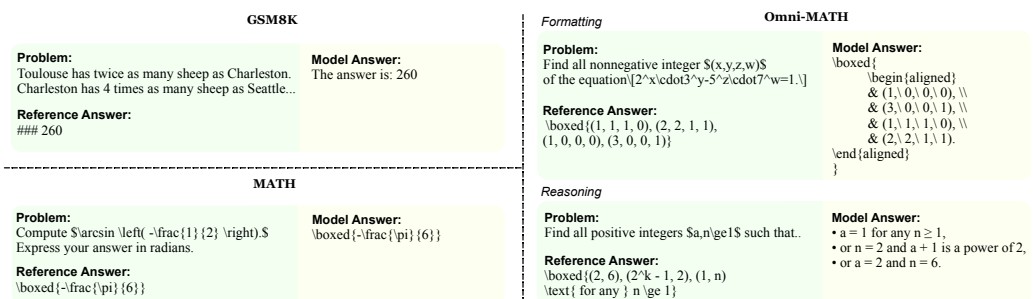

Figure 5: Examples of the problem, reference answer, and model-generated answer from different datasets, where the model-generated answers are all correct. On the right side, model-generated answers have issues in reformatting or requiring additional reasoning to correctly evaluate.

We discover that conducting answer-level assessments for Olympiad-level problems is a complex task due to the diverse formats of the final answers produced by models (Figure 5 and Section H.3). This diversity makes it challenging to evaluate model outputs using fixed rules, as seen in other benchmarks, and we accordingly employ model-based evaluation for the assessment of Omni-MATH. Specifically, we leverage GPT-4o to verify whether the content generated by the tested model aligns with the standard answer. In our prompt, we provide *problem*, *reference answer*, *model-generated solution*, querying GPT-4o to determine whether the model solution is consistent with the reference answer. Detailed prompts and model outputs can be found in the Appendix H. Additionally, in Section 4.2, we assessed the evaluation capability of GPT-4o to ensure the accuracy of our results.

To promote public accessibility for Omni-MATH, we additionally filtered a subset named *Omni-MATH-Rule* for reliable rule-based evaluation, as in Appendix K. What's more, for total set evaluation, we also developed an open-source evaluation model, named *Omni-Judge*, to assess the consistency between model solutions and reference answers at a low cost. Specifically, we first constructed a dataset for training (∼17618), validation (∼2200), and test (∼2200) based on evaluation results from GPT-4o, which have no overlaps of questions with each other. Then we treat this task as a similar flow of instruction following, where the extraction of the model answer, judgment, and detailed justification are all required to be generated by Omni-Judge. We accordingly test multiple instruct models while appending a case in the context to boost the performances. It is finally evaluated on the consistency between its judges and those from GPT-4o, which surprisingly exceeds 90%, demonstrating the strong feasibility of Omni-Judge. Detailed analysis can be found in Sec 4.2 and Appendix J.

## 3 OLYMPIAD-LEVEL MATH EVALUATION ON EXISTING LLMS

### 3.1 EXPERIMENTAL SETUP

We evaluate 15 models recognized for their strong mathematical reasoning capabilities. All prompts for these models are formatted according to the guidelines provided on their respective repositories. The detailed model information and prompt specifications can be found in Appendix F. To ensure a more accurate assessment, we employ GPT-4o to evaluate whether the model outputs are consistent with the correct answers.

We then analyze the accuracy rates across different domains and levels of difficulty. For clarity and simplicity in the presentation, we report the accuracy rates of the first-level subdomains of the domain tree. For difficulty, we categorize it into four tiers according to the difficulty tag: T1: 1-3, T2: 3-5, T3: 5-7, and T4: 7-10. Detailed accuracy statistics for each distinct difficulty level are provided in Appendix G.

## 3.2 EXPERIMENTAL RESULT

Table 3: Main Result. The results with bold text represent the best performance score. *Qwen2.5-MATH-72b-Instruct* and *OpenAI o1-mini* separately perform best in the vanilla model setting and test-time scaled model setting.

| Model | Acc | Alg. | P.Cal | Cal | Geo. | D.M. | Num. | App. | #T1 | #T2 | #T3 | #T4 |
|---|---|---|---|---|---|---|---|---|---|---|---|---|
| *Vanilla Models* | | | | | | | | | | | | |
| InternLM2-MATH-mixtral8*22B | 14.24 | 18.19 | 12.50 | 10.16 | 8.70 | 8.03 | 10.09 | 12.36 | 42.78 | 8.01 | 10.35 | 6.74 |
| DeepSeekMATH-7b-RL | 16.12 | 21.28 | 20.45 | 12.50 | 9.87 | 7.71 | 9.98 | 13.58 | 49.07 | 9.11 | 11.49 | 7.80 |
| Mathstral-7B-v0.1 | 19.13 | 23.99 | 25.00 | 13.28 | 12.19 | 10.04 | 14.58 | 16.30 | 53.07 | 10.93 | 15.29 | 11.86 |
| DeepSeek-Coder-V2-Lite-Instruct | 19.73 | 24.55 | 23.86 | 13.28 | 13.06 | 8.92 | 15.88 | 16.81 | 55.93 | 13.15 | 12.86 | 9.55 |
| MetaLlama-3.1-70B-instruct | 24.16 | 29.15 | 27.59 | 18.75 | 14.76 | 11.74 | 17.03 | 24.66 | 62.66 | 16.82 | 16.95 | 13.71 |
| DeepSeek-Coder-V2 | 25.78 | 30.24 | 35.23 | 15.62 | 17.99 | 12.71 | 20.90 | 23.58 | 65.38 | 18.84 | 18.06 | 14.61 |
| Claude-3.5-SONNET | 26.23 | 30.30 | 29.55 | 19.53 | 17.70 | 15.74 | 19.51 | 26.70 | 66.23 | 18.91 | 18.27 | 17.41 |
| NuminaMATH-72B-COT | 28.45 | 34.74 | 27.27 | 21.88 | 20.41 | 16.95 | 23.47 | 25.06 | 65.63 | 23.70 | 20.33 | 21.08 |
| Qwen2-MATH-7b-Instruct | 29.36 | 36.08 | 35.23 | 24.22 | 18.68 | 14.41 | 27.04 | 25.93 | 63.52 | 24.30 | 21.52 | 18.54 |
| GPT-4o | 30.49 | 36.12 | 39.77 | 21.88 | 21.57 | 15.74 | 25.75 | 29.38 | 68.38 | 25.01 | 21.83 | 15.81 |
| Qwen2.5-MATH-7b-Instruct | 33.22 | 39.39 | 37.50 | 31.25 | **26.89** | 16.93 | 28.62 | 30.37 | 66.23 | 29.20 | 24.68 | 20.34 |
| Qwen2-MATH-72b-Instruct | 33.68 | 40.27 | 37.50 | 27.34 | 22.53 | 17.50 | 30.01 | 32.96 | 70.10 | 29.06 | 24.71 | 17.98 |
| Qwen2.5-MATH-72b-Instruct | **36.20** | **43.33** | **42.53** | **39.84** | 26.57 | **18.28** | **34.28** | **33.37** | **70.96** | **31.37** | **27.75** | **22.29** |
| *Test-time Scaled Models* | | | | | | | | | | | | |
| Qwen2.5-MATH-7b-Instruct RM@8 | 35.70 | 42.12 | 36.78 | 33.59 | **31.89** | 18.96 | 29.59 | 30.88 | 67.95 | 31.46 | 27.41 | 24.0 |
| Qwen2.5-MATH-7b-Instruct RM@256 | 35.79 | 42.54 | **49.43** | **39.06** | 25.79 | **19.75** | 31.66 | 33.13 | 68.24 | 30.48 | **27.81** | 23.71 |
| Qwen2.5-MATH-72b-Instruct RM@8 | **36.34** | **43.89** | 48.28 | 34.38 | 26.18 | 18.28 | **33.30** | **34.12** | **71.24** | **32.04** | 26.94 | 23.43 |
| Qwen2.5-MATH-72b-Instruct RM@256 | 35.95 | 43.47 | 47.13 | 35.94 | 25.10 | 19.41 | 32.64 | 34.12 | 68.38 | 31.46 | 27.68 | **26.28** |
| OpenAI o1-preview | 52.55 | 57.70 | 57.47 | 53.91 | 43.11 | 31.26 | 49.67 | 53.42 | 80.11 | 50.83 | 42.25 | 37.71 |
| OpenAI o1-mini | **60.54** | **67.82** | **68.18** | **60.94** | **51.50** | **37.68** | **61.74** | **60.52** | **82.23** | **63.10** | **49.11** | **42.69** |

**Olympic-level mathematics remains far from solved** Our findings indicate that Omni-MATH currently presents significant challenges to all models. The strongest model evaluated is OpenAI o1-mini, which utilizes test-time enhancement to achieve an accuracy rate of just 60.54%. OpenAI o1-preview attains an accuracy of 52.55%, while the SOTA vanilla model acquires 36.2%, leaving a substantial gap from the top two models. It should be noted that open-source models, like Qwen2.5-MATH, have surpassed GPT-4o in the field of Olympiad mathematical reasoning.

**Domain-Specific Analysis** Models demonstrate a stronger proficiency in domains such as algebra, calculus, and number theory while showing significant weaknesses in discrete mathematics. We hypothesize that this phenomenon stems from the prevalence of the former subjects in mathematical datasets; nearly all datasets reference algebra and calculus. Conversely, datasets related to discrete mathematics are scarcer, rendering this domain particularly challenging for models.

**Limitations in Best-of-N Scaling** The efficient test-time scaling techniques warrant further development. We have employed the common approach in the current mathematical reasoning studies, known as Best-of-N (Wang et al., 2024; Gao et al., 2024; Yang et al., 2024b), utilizing the Qwen2.5-Math-RM-72B. However, we found that scaling the inference time did not yield consistent improvements in performance. We propose two potential reasons: (1) The reward model demonstrates a limited ability to supervise tasks related to Olympic-level mathematics. (2) The policy model struggles to search for the correct solutions. Moreover, with the limitations on inference tokens for OpenAI o1-preview and OpenAI o1-mini set to a maximum of 4096, which is considerably lower than the RM@256 costs, we still observe performance that greatly surpasses that of vanilla models. Thus, the development of more efficient test-time enhancement approaches remains an open area for exploration. The further discussion about the underlying problem is detailed in Appendix I.

# 4 ANALYSIS

## 4.1 DATA LEAKAGE ANALYSIS

To eliminate the potential impact of data contamination on the conclusions of the experiments above, it is essential to conduct data leakage detection. Following Xu et al. (2024), we employed *n-gram accuracy* to identify any data leakage present in the existing models. Specifically, for each sample in the dataset, we concatenated the problem and its corresponding solution, then randomly selected $K$ positions for 5-gram extraction. Whether the given sample has been contaminated corresponds to whether the 5 grams predicted by the model are identical to the ground truth 5 grams. We report the results in Table 4.

Table 4: Results of data contamination detection. (a) Contaminated; (b) Contaminated & Correct.

| Model | (a) | | (b) | |
|---|---|---|---|---|
| | Prop. | Avg.D. | Prop. | Avg. D. |
| DeepSeekMATH-7b-RL | 0.23% | 4.02 | 0.02% | 1.0 |
| DeepSeekCoder-V2-Lite | 0.27% | 4.25 | 0.07% | 3.2 |
| Qwen2-MATH-7b-instruct | 0.41% | 5.3 | 0.09% | 4.12 |
| Mathstral-7B-v0.1 | 0.29% | 4.02 | 0.07% | 2.16 |
| Qwen2.5-MATH-7b-instruct | 0.63% | 4.61 | 0.16% | 4.2 |
| Qwen2.5-MATH-72b-instruct | 0.70% | 5.39 | 0.27% | 4.54 |
| Qwen2-MATH-7b-instruct | 0.47% | 5.27 | 0.16% | 4.5 |
| NuminaMATH-72B-COT | 0.56% | 2.46 | 0.34% | 1.9 |
| MetaLlama-3.1-70B-instruct | 0.09% | 5.25 | 0.07% | 4.8 |
| GPT-4o | 0.04% | 4.5 | 0.04% | 4.5 |

It can be seen that most models exhibit certain data leakage, as Omni-MATH is based on data sourced from the internet, while Qwen2.5-MATH-72b-instruct has the highest degree of data leakage with 5 grams accurately predicted in 31 samples. We also collect the samples that are contaminated and then correctly answered by the tested models, finding that the numbers of the "Contaminated" and "Contaminated & Correct" samples are both extremely low. It indicates that data leakage has little impact on the conclusions before and Omni-MATH still poses significant challenges for them.

Moreover, we focus on the distribution of problem difficulty in these contaminated samples, where for most tested models, the average difficulty of "Contaminated & Correct" samples is consistently lower than that of all contaminated samples, except GPT-4o and MetaLlama-3.1-70B-instruct that almost have no data leakage. It seems that difficulty level continues to exert a notable influence on the performance of models even in scenarios of data leakage.

## 4.2 RELIABILITY OF JUDGMENT MODELS

Despite the traditional rule-based evaluation, model-based evaluation provides a flexible evaluation in an end-to-end manner without the need to extract the answers and build complex rules. There are several studies utilizing model-based evaluation like GPT-4 to evaluate the correctness of answers (Fang et al., 2024; Huang et al., 2024), but whether such judgment models can consistently provide reliable feedback on Olympic-level problems remains inconclusive. In this part, we conduct a meta-evaluation experiment to assess the reliability of model-generated judgments. In detail, we first build a subset containing 100 samples and engage several graduate and doctoral students to annotate each sample as golden judgments, i.e. whether each solution is aligned with the standard answer for the given problem, as details in Appendix C.3. Next, the predictions from judgment models are compared with the gold-standard annotations. We were surprised to find that GPT-4o acquires a 98% accuracy with human annotations, demonstrating the strong reliability of both the evaluation tool and the results in Omni-MATH. Additionally, Omni-Judge also shows considerable effectiveness with an accuracy of 86%, making it a cost-effective tool for quickly evaluating and iterating models.

Since GPT-4o shows great consistency with human annotations, we treat it as a reliable proxy of human judgment and further analyze the performance of Omni-Judge in Appendix J.

## 4.3 EMPIRICAL ANALYSIS ON DATA SELECTION

In this section, we aim to provide practical conclusions of the data selection in pursuit of a better Olympic-level mathematical reasoner. Here we develop a novel approach to estimating the impact of domain/difficulty distributions on the final performance, that is, for each model, we manipulate demonstrations in the input and observe variation in performance through in-context learning.

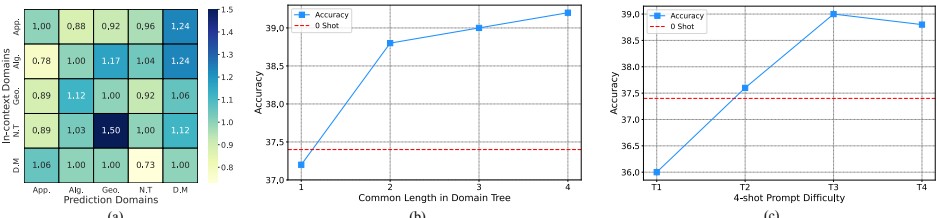

Figure 6: Results of data selection experiments. (a) The impact of different demonstration domains on model performance. (b) The effect of the distance within the domain tree on model performance. (c) The influence of the difficulty level of in-context examples on model performance.

To balance the in-context ability of the model and its mathematical reasoning ability, we choose GPT-4o as the base model for this experiment. Due to cost constraints, we randomly sample 500 data from Omni-MATH as the evaluation set for this section. The following are the details and conclusions drawn from our experiments.

**Domain**  To investigate the mutual influence of data from different domains, we retrieve data from different domains as in-context demonstrations. To balance reasoning cost and the impact of cases, we select four same-domain examples for in-context learning. We present the mutual influence among 5 main domains in Figure 6 (a). For the sake of presentation, we normalize the column according to the diagonal data, i.e., putting in-domain data as the baseline. From the experimental results, we observe that the mutual influence among these domains is asymmetric. For example, Discrete Mathematics has a worse influence on Number Theory, while Number Theory has a positive effect on the prediction of Discrete Mathematics. Additionally, Applied Mathematics and Algebra show a significant positive effect on Discrete Math, with an accuracy of 124% of the in-domain data. Also, Number Theory greatly supports Geometry, with accuracy reaching 150% of in-domain accuracy. Additionally, we find that Algebra interferes with applied mathematics.

**Common Length**  The similarity between the in-context examples and the target domain is an important factor that may influence the model's performance. To explore this, we conduct experiments to examine the impact of the fine-grained domain similarity between demonstrations and the target domain on the model's effectiveness. The results are shown in Figure 6 (b). We define domain similarity as the length of a common path on the domain tree. A longer common path indicates greater domain similarity, and we find that as the domain similarity increases, the model's performance consistency also increases. Compared to random selection, the most significant improvement is brought by the same first-level domain data, suggesting that the model has the ability to generalize the in-domain data at a broader level.

**Difficulty**  We also discussed the impact of the difficulty of in-context examples on model performance. Consistent with the settings in Sections 3.1 and 4.4, we divided the difficulty into 4 intervals. We find that as the difficulty of examples in the prompt increases, the model's performance gradually improves, except for a slight decline on T4. Overall, we can conclude that more difficult data would help improve the model performance.

## 4.4 DIFFICULTY CONSISTENCY ANALYSIS

Recent studies (Yang et al., 2024c) have shown that models may exhibit the capability to solve difficult problems, yet fail to perform well on easier ones. Specifically, the accuracy of solving challenging problems is not consistently higher than that of simpler ones in all cases. In this section, we examine the consistency of model performance within the problem difficulty in Omni-MATH.

Our difficulty tag is obtained differently from the pairwise data construction employed in Yang et al. (2024c), so it is unsuitable to directly utilize the consistency score from their study. However, we can measure the trend intensity of model performance as the difficulty increases for consistency. Specifically, we use the four difficulty intervals in Table 3 for the overall assessment. The model performance is defined by the variables $x_1, x_2, x_3, x_4$, where $0 < x_i < 100$. Then we define the

trend intensity $\mathcal{A}$ by simplifying the implementation of ARIMA Modeling (Box & Pierce, 1970) and adapting it to our situation. The implementation of $\mathcal{A}$ with $n$ variables is shown below:

$$\mathcal{A} = \sum_{i=1}^{n-1} \begin{cases} K \cdot (x_i - x_{i+1}) & \text{if } x_{i+1} > x_i \\ \min(\frac{max(x)}{n}, x_i - x_{i+1}) & \text{if } x_{i+1} <= x_i \end{cases} \quad (1)$$

This equation measures the predicted values and their relative comparison at each position based on the first difference. We assume that $x$ decreases as $i$ increases. For counter-trend situations, we apply a penalty coefficient that is double that of positive trends. We constrain the largest impact of each position to the $\frac{max(x)}{n} = 25$. For sequences of four variables with a penalty coefficient of $K = 2$, the value range of $\mathcal{A}$ is $[-200, 75]$. Intuitively, the more pronounced the downward trend of $x$, the larger the value of $\mathcal{A}$; conversely, the smaller the value of $\mathcal{A}$. The experimental result is shown in Figure 7.

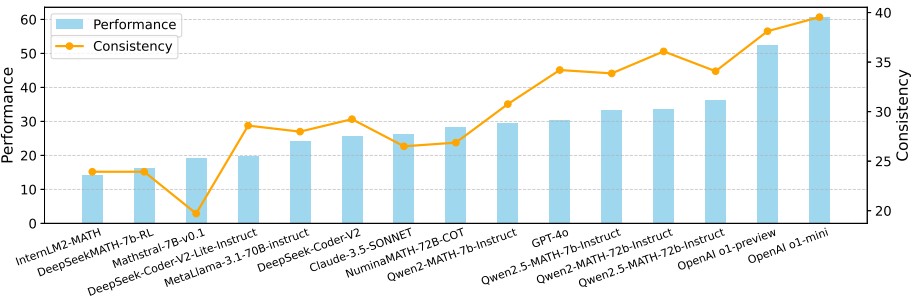

Figure 7: Performance and Consistency Comparison of Different Models. Consistency is measured by Equation 1. The difficulty consistency represents shows that our difficulty level is generally aligned with the performance of models.

From the experimental results, we observe that the consistency of all models is positive, indicating that as the difficulty increases, the overall accuracy of all models declines. This finding also validates the rationality of our difficulty classification. Additionally, we note that as model capabilities improve, their consistency also increases, but exceptions also exist. This conclusion is similar to that reached in Yang et al. (2024c), even though our methods of measurement are entirely different.

## 5 RELATED WORK

### 5.1 MATHEMATICS BENCHMARKS

Measuring the mathematical reasoning abilities of language models has long been a focal point in both academia and industry. GSM8K (Cobbe et al., 2021) and MATH (Hendrycks et al., 2021) are the key benchmarks in the mathematical reasoning domain. GSM8K primarily emphasizes a model's capacity to solve practical application-based math problems, while MATH presents a greater challenge, encompassing university-level and introductory competition-level mathematics questions. As models become more powerful, several works have proposed more challenging problems, such as OCWCourses (Lewkowycz et al., 2022), SAT (Azerbayev et al., 2024), JeeBench (Arora et al., 2023), and MATHOdyssey (Fang et al., 2024). However, as models rapidly improve, the challenge posed by these benchmarks is gradually diminishing, making it increasingly difficult to differentiate their mathematical reasoning abilities. Another line of research (Zheng et al., 2022; Azerbayev et al., 2023) employs formal languages to measure the models' theorem proving capabilities.

### 5.2 OLYMPIAD-LEVEL BENCHMARKS

To further explore the boundaries of large language model capabilities, many studies have attempted to yield Olympiad-level data as benchmarks. Among these, AlphaGeometry (Trinh et al., 2024) offers Olympiad-level geometry problems. CHAMP (Mao et al., 2024) provides high school competition-level math problems, distinctly annotated with concepts and hints relevant and helpful for each problem. OlympiadBench (He et al., 2024) and OlympicArena (Huang et al., 2024)

are two comprehensive benchmarks that encompass Olympiad problems from various fields. However, within these benchmarks, the proportion of text-only mathematical reasoning data remains scarce. Furthermore, none of the aforementioned benchmarks have further differentiated and analyzed the problems from Olympiad mathematics competitions. Our proposed Omni-MATH focuses exclusively on text-only Olympiad-level mathematical reasoning to truly explore the boundaries of current LLM's mathematical capabilities. We further categorize the problems from the Olympiad mathematics competitions into various difficulty levels, providing a clear and reasonable difficulty classification setting for better diagnosing model performance.

## 6 CONCLUSION

In this paper, we introduce Omni-MATH, a comprehensive and highly challenging benchmark for Olympiad-level mathematical reasoning. Our experiments reveal that even the most advanced model in mathematical reasoning, OpenAI o1-mini, achieves a maximum accuracy of only 60.5%, while the open-source model currently achieves only 36.2%. These results demonstrate that the highest difficulty level in mathematics presents significant challenges for large models. Additionally, we conducted an in-depth analysis of model performance based on this dataset, aiming to provide valuable contributions to the mathematics community.

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

# A    ANALYSIS OF PROCESS-LEVEL ASSESSMENT

In this section, we utilize GPT-4o for process-level assessment. Specifically, we provide GPT-4o of the question, the corresponding standard answer, and the model-generated answer in the prompt. We split both the reference answer and model-generated answer into steps and let GPT-4o evaluate the correctness of each step of the model-generated answer. If a step is identified as incorrect, we further query an explanation for the error. Following the Gao et al. (2024), we categorize the mathematical reasoning steps as follows: **Unrelated**: This indicates that the step is irrelevant and does not contribute towards deducing the final answer. **Accumulation**: This denotes that the step is incorrect due to a mistake in the preceding step, leading to subsequent errors. **Calculation**: This categorization is reserved for errors arising from incorrect calculations, which is one of the most common errors in mathematical reasoning. **Logic**: This applies to steps that are logically flawed in the context of solving the given problem. **Other**: This category encompasses steps that are erroneous for reasons not covered by the aforementioned categories. Then we compiled the proportions of step-level error types and the model performances, as illustrated in Figure 8.

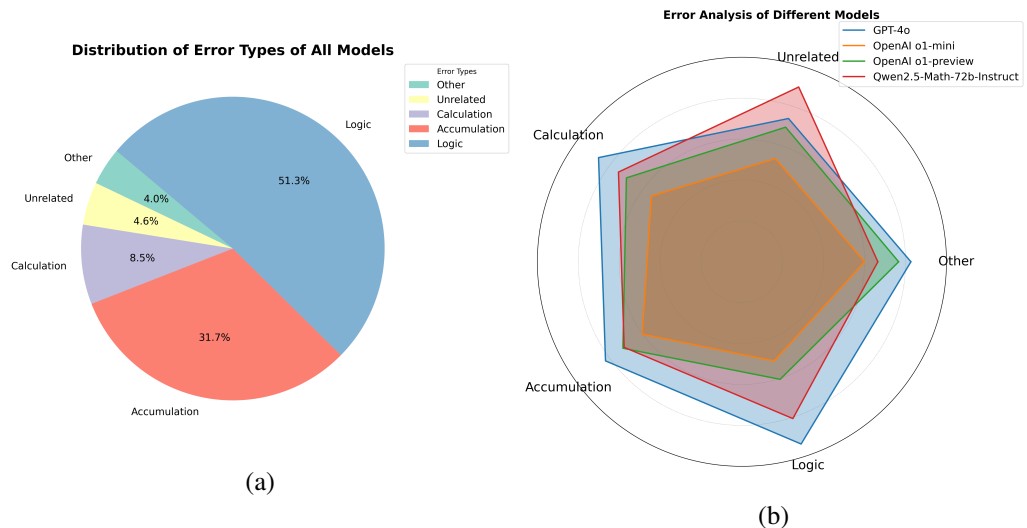

(a)

(b)

Figure 8: Process-level result across differnec

**Logical errors represent a significant issue for the models.** As shown in Figure 8 (a), among the best existing inference models, most erroneous steps can be attributed to logical errors, followed by accumulation errors and calculation errors. This highlights the challenges posed by our proposed assessment set for current models.

**OpenAI O1-mini demonstrates significantly fewer errors across all categories.** Figure 8 (b) illustrates the error distribution across different categories for various models. Given that the total number of errors varies by category, we applied normalization. Our findings indicate that the OpenAI O1-mini model outperforms other models across all error categories. Notably, GPT-4o exhibits a higher incidence of calculation and logical errors compared to other math-specialized models, underscoring that specialized training in mathematics is essential for Olympiad-Level mathematical reasoning. Additionally, Qwen2.5-MATH-72b-Instruct shows a tendency to produce unrelated content compared with other models.

# B  DETAILED DATA LEAKAGE INFORMATION

We examine whether the given instance has been contaminated by testing whether the model's predicted 5 grams are identical to the ground truth 5 grams. The detailed experimental results are illustrated in the Figure 9.

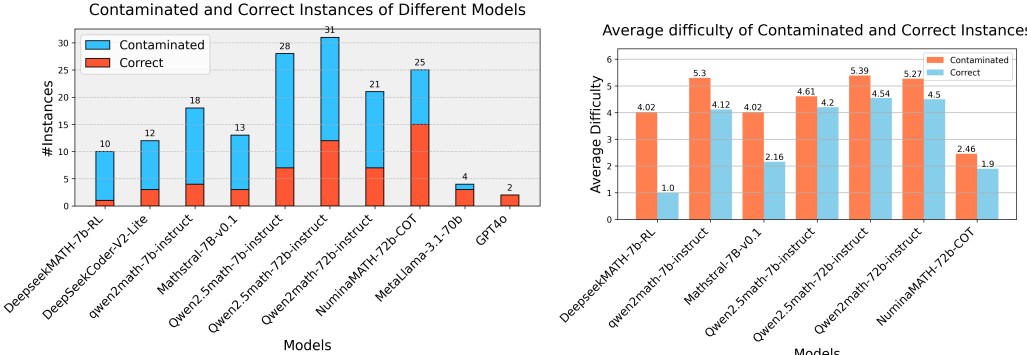

(a) Number of instances of different models affected by contamination and the correct instances under contaminated conditions.

(b) Average difficulty of contaminated and correct instances.

Figure 9: Data Leakage and Correct Instances for Different Models.

# C  ANNOTATION DETAILS

## C.1  ANNOTATION GUIDELINE

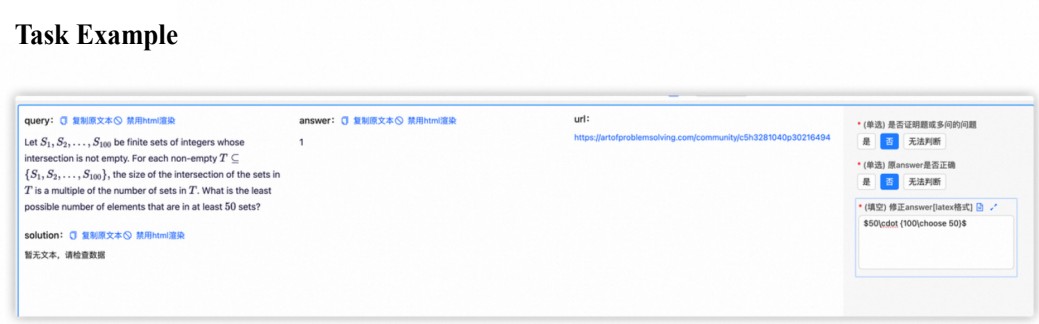

Figure 10: The illustration of the UI page of the annotation task.

The detailed annotation tasks and principles are listed here. Specifically, the annotators were assigned the following three tasks:

- Determine whether the current problem contains multiple questions or is proof-based. If it is a multi-part question or a proof question, the evaluation of this data becomes more complex, making it unsuitable for the evaluation in our benchmark; therefore, we decided to discard this part of the data.

- Verify the correctness of the answers present in the existing dataset. This step is essential for assessing answer quality; different verification methods are required depending on the data source. For entries from AoPS Wiki and problems with provided solutions, it suffices

to check for consistency with the answers in the data source. For data from the AoPS Forum, it is also necessary to refer back to the results provided in the forum.

- If the original answer is found to be incorrect, provide a corrected answer based on the data source.

## C.2 ANNOTATION DETAILS

For the 1100 questions collected from the AoPS Forum, we employed 4 annotators to verify the consistency between the final responses and the original forum replies. Each annotator labeled 550 questions, resulting in two annotations per question. We then conducted cross-validation and sampled cases for inspection. The accuracy of the cross-validation was 92.7%(1020/1100), while the post-sampling accuracy was 97.3%(146/150). After removing discrepancies, we obtained a total of 888 questions from this set.

For the Contest Page questions converted from PDF to LaTeX, the quality was exceptionally high. We employed rough filtering rules to eliminate some errors that occurred during the conversion process. Subsequently, a manual check was performed to ensure the consistency between answers and explanations. This part resulted in a total of 3295 questions.

Regarding the questions from the AoPS Wiki, comprehensive explanations were readily available. We only needed to filter out the video-based explanations and exclude questions that did not fit within our collection scope, such as AMC and AIME questions. This section yielded a total of 298 questions.

After the labeling process, the initial dataset comprised 4481 questions. Subsequently, we hired 2 annotators to inspect the entire dataset, following a set of principles. After a final filtration, 53 questions were removed from the Contest Page source, resulting in a final dataset of 4428 questions.

## C.3 DETAILS OF META-EVALUATION SET

The 100 samples of the meta-evaluation set mentioned in Section 4.2 were derived from the inference results of DeepSeek-Coder-V2, GPT-4o, and Qwen2.5-MATH-72b-Instruct, sampled according to difficulty to ensure representation across all levels of difficulty without overlapping questions. Although the inference results come from different models, our ultimate evaluation target is the human alignment of judgment models—GPT-4o and Omni-Judge. Therefore, all 100 examples are designed for GPT-4o or Omni-Judge to provide reliable human evaluation.

Our experiments revealed that GPT-4o achieved highly accurate results under conditions of high agreement among human annotators (97% inter-agreement) given the solution generated by 3 different models. For annotation, we employed two PhD students and two Master's students, with each annotator labeling 50 problems. Moreover, they were instructed to determine whether the model's answers were correct based mostly on the correct answers rather than their own judgments. Each annotator was responsible for 50 questions, allowing us to ensure that each question was annotated by two annotators, facilitating cross-validation to assess inter-agreement. During individual annotation, three out of 100 instances of inconsistent results were observed; however, after discussion, the annotators reached a consensus on the answers. The discrepancies primarily stemmed from GPT-4o misinterpreting the "Student Answer" as the "Ground Truth" under the current prompt, which we attribute to prompt formatting issues. This problem was resolved by adjusting the few-shot prompt format.

## D  DATA CRAWLING DETAILS

To enhance the possibility of obtaining correct answers in *AoPS Wiki*, we specifically targeted posts that contained hidden tags indicating "solution", as well as those that included "\boxed{}" or "■". Following LI et al. (2024), we assume that these posts have a higher probability of containing accurate answers. Then employed GPT-4 to reformat the user-uploaded solutions into the standardized format of `solution + \boxed{Final Answer}` to improve the answer format quality.

# E    DETAILED DATA INFORMATION OF OMNI-MATH

## E.1    DETAILED DATA SOURCE OF OMNI-MATH

Table 5: The hierarchical data source of the Omni-MATH.

| Contests | Number of Problems |
|---|---|
| *Highly Influential International Competitions* | |
| IMO | 75 |
| IMO Shortlist | 190 |
| IMO Longlist | 43 |
| Putnam | 101 |
| IMC | 68 |
| *Notable International/National Challenging Competitions* | |
| USAMO | 133 |
| IZhO | 14 |
| China TST | 106 |
| CMO | 38 |
| USA TST | 45 |
| *Intermediate International/National Competitions* | |
| APMO | 85 |
| USAJMO | 39 |
| Balkan MO | 26 |
| Balkan MO Shortlist | 35 |
| JBMO | 23 |
| JBMO Shortlits | 33 |
| ToT | 55 |
| BalticWay | 37 |
| Alibaba global Contest | 22 |
| Middle European Mathematical Olympiad | 21 |
| Other Olympiads | 269 |
| *National Championships or Competitions* | |
| HMMT_2 | 1385 |
| HMMT_11 | 896 |
| Yau Contest | 7 |
| *Introductory Competitions* | |
| Pascal | 249 |
| Cayley | 201 |
| Fermat | 232 |

## E.2    DIFFICULTY

The detailed difficulty distribution of the Omni-MATH is shown in Figure 11. The difficulty distribution roughly follows a normal distribution

## E.3    DOMAIN CLASSIFICATION TREE

The overall domain tree with the problem number is shown in Figure 12. Note that the total number is larger than the total data of 4428 because the question might belong to multiple domains.

## E.4    DIFFICULTY CLASSIFICATION PROMPTS

For data not included in *Aops:Ratings*, we need to assign a difficulty score to the problem. We collect all the problems with its solution and the data source. Then we use GPT-4o to assign a difficulty level of the problem, the prompt we use is shown in Figure 13. The relevant page information makes the prompt too long to show on this page. Instead, we upload the total prompt as the additional materials.

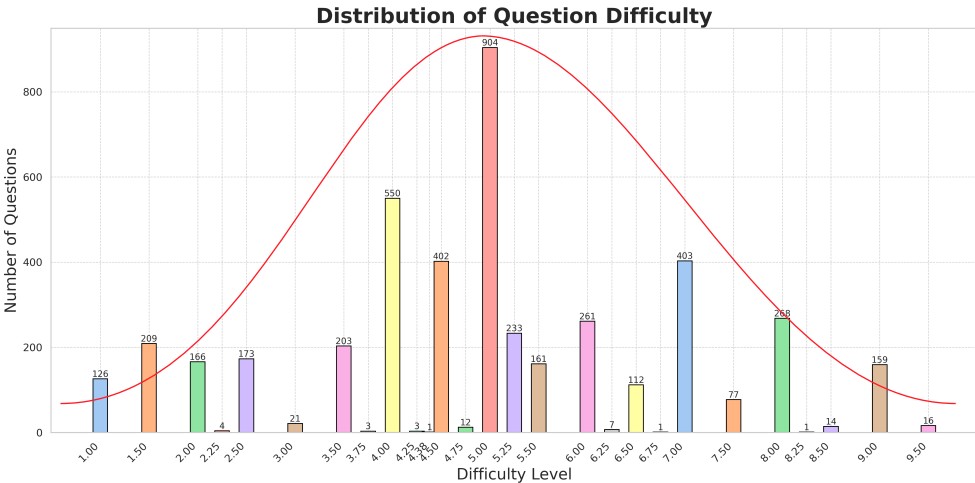

Figure 11: The difficulty distribution of Omni-MATH.

## E.5    DOMAIN CLASSIFICATION PROMPTS

We use GPT-4o to assign a domain tag to each problem, the prompt we use is shown in Figure 14. The domain tree makes the prompt too long to show on this page. Instead, we upload the total prompt as the additional materials.

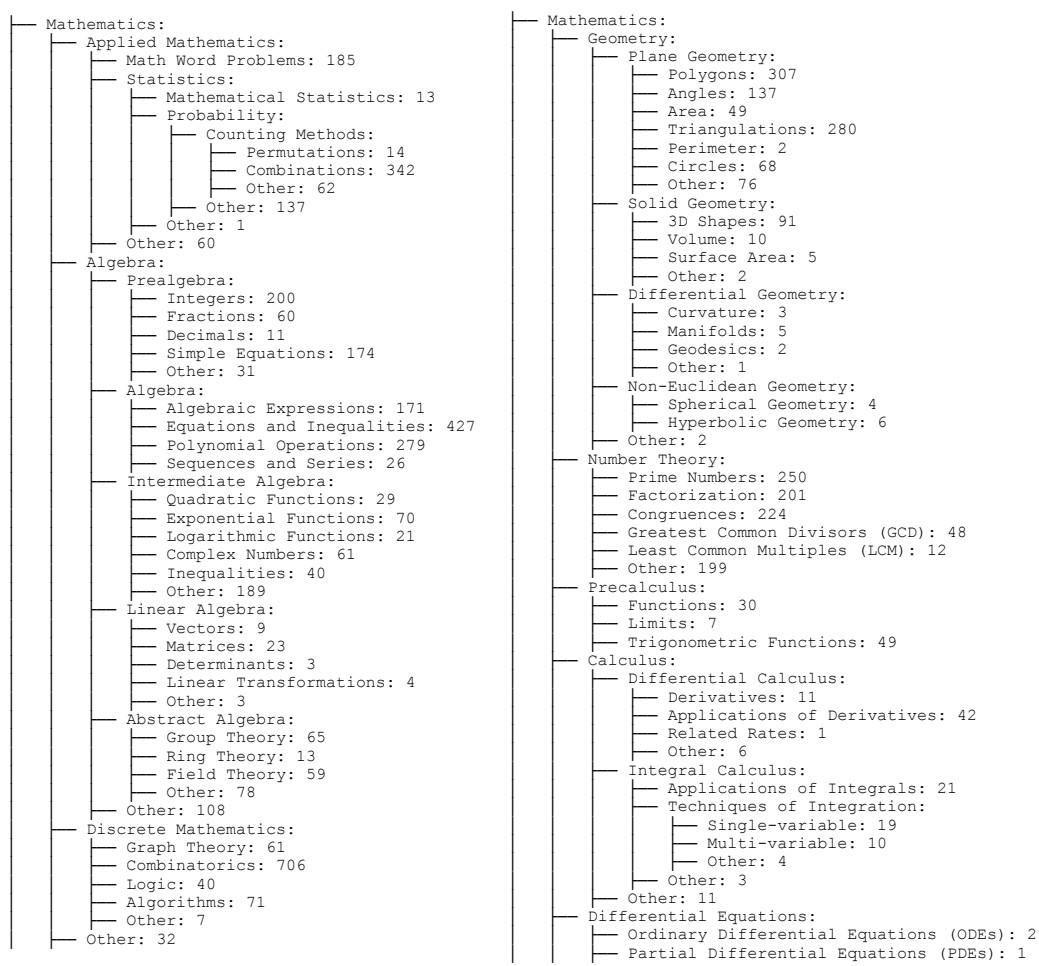

Figure 12: The total domain tree of Omni-MATH.

# F DETAILED EXPERIMENTAL SETUP

## F.1 MODELS

Here are the baseline models we evaluated:

InternLM2-MATH-mixtral8*22B (Ying et al., 2024); DeepSeekMATH-7b-RL (Shao et al., 2024); Mathstral-7B-v0.1 (AI, 2024); DeepSeek-Coder-V2-Lite-Instruct (DeepSeek-AI et al., 2024); MetaLlama-3.1-70B-instruct (Abhimanyu Dubey & Abhishek Kadian, 2024); DeepSeek-Coder-V2 (DeepSeek-AI et al., 2024); Claude-3.5-SONNET (Anthropic, 2024); NuminaMATH-72B-COT (LI et al., 2024); Qwen2-MATH-7b-Instruct (Yang et al., 2024a); Qwen2.5-MATH-7b-Instruct (Yang et al., 2024a); Qwen2-MATH-72b-Instruct (Yang et al., 2024a); Qwen2.5-MATH-72b-Instruct (Yang et al., 2024a); GPT-4o (OpenAI, 2023); OpenAI o1-preview (OpenAI, 2024); OpenAI o1-mini (OpenAI, 2024).

## F.2 MODEL INFERENCE SETUP

For all the models under evaluation, the prompts follow the instructions provided on their respective release pages, such as DeepseekMATH-7b-RL and Qwen2.5-MATH-7b.

```
# CONTEXT #
I am a teacher, and I have some high-level olympiad math problems.
I want to evaluate the difficulty of these math problems. There are some references available regarding the difficulty of the problems:
<difficulty reference>
## Examples for difficulty levels
For reference, here are problems from each of the difficulty levels 1-10:
## Some known difficulty ratings of the competitions.
(Too long to show...)
</difficulty reference>

# OBJECTIVE #
A. Summarize the math problem in a brief sentence, describing the concepts involved in the math problem.
B. Based on the source of the given problem, as well as the difficulty of the problems referenced in these materials and the solution to the current problem, please provide
an overall difficulty score for the current problem. The score should be a number between 1 and 10, with increments of 0.5, and should align perfectly with the materials.

# STYLE #
Data report.

# TONE #
Professional, scientific.

# AUDIENCE #
Students. Enable them to better understand the difficulty of the math problems.

# RESPONSE: MARKDOWN REPORT #
## Summarization
[Summarize the math problem in a brief paragraph.]
## Difficulty
[Rate the difficulty of the math problem and give the reason.]

# ATTENTION #
 - Add "=== report over ===" at the end of the report.

<example math problem>
(Too long to show)
</example math problem>

## Summarization
The problem requires finding a value that makes the equation $\\frac{1}{9}+\\frac{1}{18}=\\frac{1}{\\square}$.
This involves adding two fractions and determining the equivalent fraction.

## Difficulty
Rating: 1
Reason: This problem is straightforward and primarily involves basic fraction addition, making it suitable for early middle school students.

=== report over ===

</example math problem>
(Too long to show)
</example math problem>

## Summarization
The problem asks for the possible values of $n$ for a regular n-sided polygon that can be completely triangulated into isosceles triangles using non-intersecting diagonals.
The solution involves analyzing the properties of the diagonals forming isosceles triangles and deducing that $n$ can be expressed in terms of powers of 2.

## Difficulty
Rating: 7
Reason: The problem involves understanding properties of isosceles triangles in the context of polygon triangulation and requires critical reasoning to establish
relationships between the number of sides and powers of 2, making it more complex than typical undergraduate-level problems.

=== report over ===

<math problem>
[Question]:
{{Question Here}}
[Solution]:
{{Solution Here}}
[Source]:
{{Source Here}}
</math problem>
```

Figure 13: The detailed difficulty classification prompt.

In chat models where there are no specific instructions on how to phrase the mathematical reasoning prompts, we directly input the model with the problem along with a system prompt: "You are an experienced educator in the field of MATHEMATICS."

To mitigate randomness in the responses, we set the parameters as follows: temperature = 0, top_p = 1, and a maximum of 2048 tokens. For the O1-preview and O1-mini, due to constraints of inference costs, we configured the maximum completion tokens to 4096. Additionally, for inference with non-API models, we employed the vllm framework.

```
# CONTEXT #
I am a teacher, and I have some high-level olympiad math problems.
I want to categorize the domain of these math problems.

# OBJECTIVE #
A. Summarize the math problem in a brief sentence, describing the concepts involved in the math problem.
B. Categorize the math problem into specific mathematical domains. Please provide a classification chain, for example: Applied Mathematics -> Probability -> Combinations.
The following is a basic classification framework in the field of mathematics.
<math domains>
(Too long to show, domain tree here..)
</math domains>

# STYLE #
Data report.

# TONE #
Professional, scientific.

# AUDIENCE #
Students. Enable them to better understand the domain of the problems.
# RESPONSE: MARKDOWN REPORT #
## Summarization
[Summarize the math problem in a brief paragraph.]
## Math domains
[Categorize the math problem into specific mathematical domains, including major domains and subdomains.]

# ATTENTION #
 - The math problem can be categorized into multiple domains, but no more than three. Separate the classification chains with semicolons(;).
 - Your classification MUST fall under one of the aforementioned subfields; if it really does not fit, please add "Other" to the corresponding branch. For example: Algebra ->
Intermediate Algebra -> Other. Only the LAST NODE is allowed to be "Other"; the preceding nodes must strictly conform to the existing framework.
 - The math domain must conform to a format of classification chain, like "Applied Mathematics -> Probability -> Combinations".
 - Add "=== report over ===" at the end of the report.

<example math problem>
(Too long to show)
</example math problem>

## Summarization
The problem requires finding a value that makes the equation $\\frac{1}{9}+\\frac{1}{18}=\\frac{1}{\\square}$.
This involves adding two fractions and determining the equivalent fraction.

## Math domains
Mathematics -> Algebra -> Prealgebra -> Fractions;

=== report over ===

</example math problem>
(Too long to show)
</example math problem>

## Summarization
The problem asks for the possible values of $n$ for a regular n-sided polygon that can be completely triangulated into isosceles triangles using non-intersecting diagonals. The
solution involves analyzing the properties of the diagonals forming isosceles triangles and deducing that $n$ can be expressed in terms of powers of 2.

=== report over ===

<math problem>
[Question]:
{{Question Here}}
[Solution]:
{{Solution Here}}
[Source]:
{{Source Here}}
</math problem>
```

Figure 14: The detailed domain classification prompt.

# G   METRICS WITH DISTINCT DIFFICULTY LEVEL

In this section, we present the fine-grained performance of the model across varying difficulty levels in Table 6. Due to the limited number of instances at certain decimal levels, such as 4.75 and 2.25, we categorize all difficulty levels into integer buckets. For example, a score of 4.75 falls under the "#D5" category, while 2.25 falls under "#D3."

The results of our experiments are depicted in the figure below. We observed a consistent decline in the model's performance as difficulty increased, except for the "#D10" category, which contains very few instances (only 16), leading to significant fluctuations in results.

Additionally, we note that the accuracy for nearly all model responses in the "#D1" category is exceptionally high, indicating that this difficulty level does not effectively differentiate model performance. However, as the difficulty increases, the differences among the model performances begin to emerge. Overall, we find that the difficulty of the GSM8K dataset mainly corresponds to the difficulty of "#D1," while the MATH dataset aligns with the difficulty of "#D2." This further underscores that our proposed benchmark significantly exceeds the existing evaluation datasets.

Table 6: Main Result with different distinct difficulty levels.

| Model | Acc | #D1 | #D2 | #D3 | #D4 | #D5 | #D6 | #D7 | #D8 | #D9 | #D10 |
|---|---|---|---|---|---|---|---|---|---|---|---|
| *Vanilla Models* | | | | | | | | | | | |
| DeepSeekMATH-7b-RL | 16.12 | 82.53 | 49.06 | 27.77 | 11.83 | 7.37 | 8.51 | 12.07 | 12.61 | 8.58 | 0.00 |
| MetaLlama-3.1-70B-instruct | 24.16 | 84.12 | 65.86 | 42.92 | 20.05 | 14.99 | 16.38 | 16.01 | 16.00 | 13.75 | 12.5 |
| DeepSeek-Coder-V2 | 25.78 | 88.09 | 69.06 | 43.93 | 25.23 | 15.28 | 14.39 | 20.94 | 16.56 | 15.95 | 0.0 |
| Claude-3.5-SONNET | 26.23 | 88.88 | 69.06 | 46.46 | 26.23 | 14.59 | 15.47 | 19.20 | 17.48 | 19.01 | 6.25 |
| NuminaMATH-72B-COT | 28.45 | 84.12 | 68.26 | 45.45 | 26.02 | 19.58 | 17.64 | 23.16 | 19.93 | 23.92 | 12.5 |
| GPT-4o | 30.49 | 88.09 | 70.93 | 51.01 | 32.53 | 20.73 | 18.29 | 23.12 | 21.77 | 16.04 | 12.5 |
| Qwen2.5-MATH-7b-Instruct | 33.22 | 86.50 | 66.66 | 52.52 | 36.12 | 25.34 | 23.06 | 25.00 | 23.00 | 20.37 | 18.75 |
| Qwen2.5-MATH-72b-Instruct | 36.20 | 87.30 | 73.6 | 55.55 | 38.89 | 27.19 | 26.98 | 28.33 | 24.00 | 22.5 | 25.0 |
| *Test-time Scaled Models* | | | | | | | | | | | |
| Qwen2.5-MATH-7b-Instruct RM@8 | 35.70 | 86.50 | 69.06 | 54.04 | 37.14 | 28.20 | 26.67 | 27.92 | 24.92 | 23.12 | 25.0 |
| Qwen2.5-MATH-7b-Instruct RM@256 | 35.79 | 85.71 | 73.06 | 58.58 | 37.81 | 26.41 | 27.76 | 28.33 | 24.30 | 20.62 | 18.75 |
| Qwen2.5-MATH-72b-Instruct RM@8 | 36.34 | 84.92 | 73.06 | 59.09 | 40.51 | 27.50 | 24.80 | 27.31 | 26.15 | 24.37 | 12.5 |
| Qwen2.5-MATH-72b-Instruct RM@256 | 35.95 | 84.92 | 70.4 | 54.04 | 39.83 | 26.80 | 25.89 | 29.15 | 24.92 | 27.5 | 18.75 |
| OpenAI o1-preview | 52.55 | 88.09 | 79.2 | 76.77 | 58.55 | 46.78 | 43.21 | 41.07 | 39.38 | 40.00 | 12.5 |
| OpenAI o1-mini | 60.54 | 88.09 | 81.55 | 79.79 | 72.70 | 57.98 | 51.08 | 47.03 | 44.78 | 45.39 | 12.5 |

# H   DETAILED EVALUATION INFORMATION

## H.1   GPT-4O EVALUATION PROMPTS

Figure 15 shows the detailed prompt we use in GPT-4o based evaluation. However, the few-shot prompt is too long to show in the paper. Instead, we will provide the total prompts in our additional materials.

\# CONTEXT \#
I am a teacher, and I have some high-level math problems. I am tasked with evaluating the correctness of a student's answer.
Below, I am provided with a problem and a reference answer. Additionally, a student's answer is provided. My job is to assess whether the student's answer captures the same meaning as the reference answer, even when expressed with different wording or format.

\# OBJECTIVE \#
I need you to judge whether the student's answer is correct given the ground truth answer.

Your tasks include:
A. Identify Mathematical or Notational Equivalence: Pay special attention to any LaTeX expressions in both answers. Confirm that the mathematical relationships, variables, and operations conveyed are equivalent.
B. Provide a Justification: Conclude with a brief explanation as to why you believe the student's output is correct or incorrect, highlighting any key differences in meaning or content.

\# STYLE \#
Teaching report.

\# TONE \#
Professional, scientific.

\# AUDIENCE \#
Students. Enable them to better understand whether the answer they produce is correct.

\# RESPONSE: MARKDOWN REPORT \#
\#\# Student Final Answer
[Extract the student's final answer, which is enclosed in "\\boxed{}".]
\#\# Equivalence Judgement
[Whether the student's answer share the same meaning with the reference answer. (TRUE or FALSE)]
\#\# Justification
[Conclude with a brief explanation as to why you believe the student's answer is correct or incorrect.]

\# ATTENTION \#
  - The reference answer is ALWAYS correct. You should carefully judge whether the student gives the same answer as reference answer.
  - The Equivalence Judgement is only TRUE or FALSE. The answer is FALSE even if the student's final answer almost correct with a minor mistakes.
  - The answer is contained within the "boxed" section, so you can focus solely on comparing the content in the student's answer box with the reference answer, without needing to consider the intermediate steps.
  - Add "=== report over ===" at the end of the report.

<example math solution>
(Too long.... We Omit this)
</example math solution>

\#\# Student Final Answer
216

\#\# Equivalence Judgement
FALSE

\#\# Justification
The student's answer of 216 is incorrect in the context of the problem, which asks for the total count of 4-digit numbers beginning with 1 that have exactly two identical digits. The reference answer is 432.
In the student's solution, they consider only cases where the digit '1' is one of the identical digits. However, the problem also includes the scenario where the identical digits could be different from '1'.
Thus, the student's calculation does not account for all valid configurations. The discrepancy in figures indicates that the student's answer does not share the same meaning as the reference answer.

=== report over ===

<example math solution>
(Too long.... We Omit this)
</example math solution>

\#\# Student Final Answer
\text{Susana has a winning strategy.}

\#\# Equivalence Judgement
FALSE

\#\# Justification
The student's answer states that "Susana has a winning strategy," which is not in agreement with the reference answer indicating "No winning strategy; the game can last indefinitely." The essence of the problem is that neither player can guarantee a win because the game can go on indefinitely without either player being able to write a polynomial with an integer root. The student's conclusion contradicts this, suggesting that a specific winning strategy exists for Susana, which is fundamentally incorrect according to the problem's conditions. Thus, the student's answer does not capture the same meaning as the reference answer.

=== report over ===

<example math solution>
(Too long.... We Omit this)
</example math solution>

\#\# Student Final Answer
None

\#\# Equivalence Judgement
FALSE

\#\# Justification
The student's answer does not provide a final conclusion and the final answer is missing, which indicates a lack of conclusion.

=== report over ===

<math solution>
\*\*Question\*\*:
{{Problem}}

\*\*Reference Answer\*\*
{{Reference Answer}}

\*\*Student Solution\*\*:
{{Solution}}

</math solution>

Figure 15: The evaluation prompt of GPT-4o.

## H.2 CASE STUDY OF EVALUATION ISSUE

---

**Problem:**
Find all positive integer $ m$ if there exists prime number $ p$ such that $ n^m\minus{}m$ can not be divided by $ p$ for any integer $ n$.

**Reference Answer:**
m \neq 1

**Model-generated Solution:**
To solve the problem, we need to determine all positive integers \\( m \\) for which thereexists a prime number \\( p \\) such that \\( p \\) does **not** divide ....
###Answer: **All positive integers m at least two. Every integer m $\geq 2$ satisfies the stated condition.**

**GPT-4o-Evaluation:**
Equivalence Judgement: TRUE,
Justification: The student's answer indicates that all positive integers \\( m \\geq 2 \\) satisfy the given condition, which is mathematically equivalent to the reference answer \\( m \\neq 1 \\). Both answers correctly identify that \\( m = 1 \\) does not satisfy the condition, while all other positive integers do. Thus, the meaning conveyed by the student's answer aligns with the reference answer.

---

**Problem:**
Find all nonnegative integer solutions $(x,y,z,w)$ of the equation\\[2^x\cdot3^y-5^z\cdot7^w=1.\\]

**Reference Answer:**
(1, 1, 1, 0), (2, 2, 1, 1), (1, 0, 0, 0), (3, 0, 0, 1)

**Model-generated Solution:**
To find all nonnegative integer solutions \\((x, y, z, w)\\) to the equation:.......
###Answer: \[
       \boxed{
              \begin{aligned}
              & (1,\ 0,\ 0,\ 0), \\
              & (3,\ 0,\ 0,\ 1), \\
              & (1,\ 1,\ 1,\ 0), \\
              & (2,\ 2,\ 1,\ 1).
              \end{aligned}
              }
\]

**GPT-4o-Evaluation:**
Equivalence Judgement: TRUE,
Justification: The student's final answer, which consists of the solutions \\((1, 0, 0, 0)\\), \\((3, 0, 0, 1)\\), \\((1, 1, 1, 0)\\), and \\((2, 2, 1, 1)\\), matches exactly with the reference answer. Each tuple denotes a nonnegative integer solution to the given equation, \\[2^x\cdot3^y-5^z\cdot7^w=1.\\] The correspondence of each part of the solution shows that the student's solution is mathematically equivalent and correctly identifies all potential solutions.

---

Figure 16: The case study of GPT-4o-based evalution.

In Figure 16, we selected data from two Human Annotated Datasets that represent two scenarios where rule-based evaluation is impractical. However, GPT-4o demonstrates the capability to assess correctly in both situations.

The first scenario is *reasoning*. It requires a nuanced understanding of the problem statement, which implies that $m$ is a positive integer greater than 0. The correct answer is $m \neq 1$; however, the model generates the answer as $m \geq 2$. Considering the defined domain for $m$ and the constraints of the answer, we can infer that these two expressions are equivalent. Thus, a certain level of reasoning ability is necessary to arrive at the correct conclusion.

The second scenario involves *complex formatting*. In this case, the order of the answers does not align with the sequence of the tuples generated by the model in LaTeX, leading to inconsistencies. This makes it challenging to employ rule-based evaluations effectively.

Table 7: Distributions of answer formats in 200 problems randomly sampled from Omni-MATH.

| Number | LaTex | Tuple | Multi-LaTeX | Function | Multi-Function | Text |
|--------|-------|-------|-------------|----------|----------------|------|
| 95 | 51 | 2 | 8 | 9 | 1 | 34 |

### H.3 DISTRIBUTIONS OF ANSWER TYPES

Rule-based answer extraction and evaluation were typical methods in LLM assessment of mathematical capacity, as utilized in prior works like MATH Hendrycks et al. (2021); He et al. (2024); Huang et al. (2024). Nevertheless, real-world mathematical problems vary widely in domain and corresponding **answer format**. To better assess the capabilities of LLMs as real-world problem solvers, Omni-MATH goes beyond simple types of answer formats like multiple-choice or direct number, incorporating more problems with diverse formats.

In this part, we conduct an analysis from this perspective with randomly sampled 200 problems, as shown in Table 7, where the answer formats are split into several types:

- **Number**    A numeric answer.
- **LaTex**    A LaTeX formula.
- **Tuple**    A tuple-formatted answer.
- **Multi-LaTex**    A combination of multiple LaTeX formulas.
- **Function**    A custom-defined function.
- **Multi-Function**    A combination of multiple custom-defined functions.
- **Text**:    A free-form explanation.

From the statistics, problems formatted in **Number**, **LaTex** and **Tuple**, which can easily extracted with rules, account for only 74% of the total, while problems with more complex answers also make up a significant portion, suggesting that results across these problems is crucial. To this end, the GPT-4 evaluation applied in Omni-MATH can be useful and accurate for it effectively covers different formats and achieves an impressive 98% consistency with human annotation (see Sec 4.2). It ensures that Omni-MATH offers a thorough and fair assessment of mathematical capacity.

## I    FURTHER DISCUSSION ON TEST-TIME SCALING RESULTS

Table 3 indicates that RM@256 fails to perform better than RM@8 on Qwen2.5-MATH-72b-Instruct, which seems counterintuitive. We find that it can be more likely attributed to the fact that the RM involved has insufficient supervision over Olympiad-level reasoning tasks than policy models. Specifically, it cannot accurately select the correct reasoning path from numerous candidates. Here we demonstrate it with the following test.

We measured the performance of the policy model: Qwen2.5-MATH-Instruct across N samples. If at least one reasoning trace is correct, we consider the example to assess the interference in RM selection, we also measured the proportion of correct COTs within the passing examples. Considering the inference cost, we sampled a subset of 500 instances according to difficulty to ensure that the distribution of problem difficulty is similar to the original dataset. The experimental result is shown below:

Table 8: The performance of the policy model under Pass@K and the proportion of correct COT.

| Metric | Pass@1 | Pass@8 | Pass@16 | Pass@32 |
|---|---|---|---|---|
| Acc | 40.8% | 57.6% | 63% | 67.2% |
| Correct COT Proportion | 100% | 68.5% | 62.5% | 58.5% |

The results show that as the inference sampling increases, the policy model is capable of solving the majority of the problems. However, even after 32 samples, 33.8% of the problems still do not yield the correct solution. The second row of the table illustrates that as the number of samples increases, the proportion of interference items also increases. In the case of 32 samples, the correct answer proportion of the passed cases drops from 100% to 58.5%. This indicates that as the sampling count increases, it becomes increasingly challenging for the reward model to select the correct COT.

Table 9: Omni-Judge results. We implement Omni-Judge based on three base models: LLaMA-2-7b-Chat, LLaMA-3-8b-Instruct, and LLaMA-3.1-8b-Instruct, and evaluate their performance according to the rate of successfully parsing outputs and the rate of consistency between the judge of Omni-Judge and GPT-4o for each problem.

| Model | Training with Homology Data | LLaMA-2-7b-Chat | | LLaMA-3-8b-Instruct | | LLaMA-3.1-8b-Instruct | |
|---|---|---|---|---|---|---|---|
| | | Success | Consistency | Success | Consistency | Success | Consistency |
| MetaLlama-3.1-70B-instruct | ✓ | 98.81 | 73.63 | 99.76 | 77.67 | 99.76 | 82.19 |
| DeepSeek-Coder-V2 | ✓ | 97.56 | 38.36 | 100.00 | 93.35 | 100.00 | 94.01 |
| Qwen2.5-MATH-7b-Instruct | ✓ | 98.45 | 35.92 | 99.78 | 89.80 | 100.00 | 90.69 |
| OpenAI o1-preview | ✓ | 99.33 | 55.03 | 100.00 | 90.83 | 99.78 | 91.28 |
| OpenAI o1-mini | ✓ | 99.78 | 63.11 | 100.00 | 89.56 | 100.00 | 91.78 |
| Mathstral-7B-v0.1 | ✗ | 99.33 | 31.49 | 99.78 | 95.12 | 100.00 | 95.79 |
| NuminaMATH-72B-COT | ✗ | 100.00 | 31.11 | 100.00 | 88.89 | 100.00 | 90.44 |
| Qwen2.5-MATH-72b-Instruct | ✗ | 98.66 | 37.28 | 99.78 | 91.96 | 100.00 | 93.30 |
| Total | - | 98.99 | 45.50 | 99.89 | 89.75 | 99.94 | 91.26 |

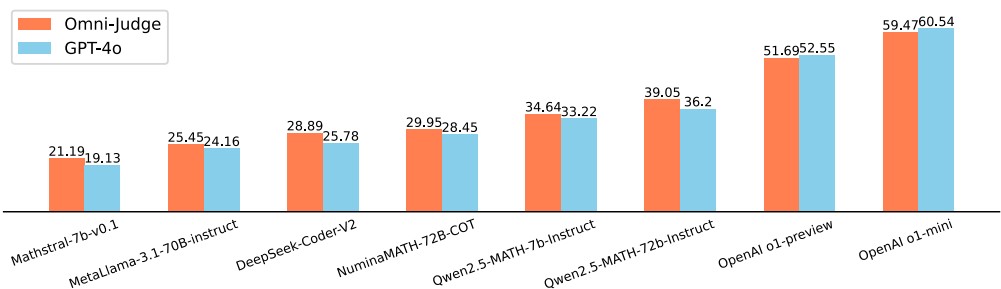

Figure 17: Results of predicted accuracy on different models, conducted by Omni-Judge and GPT-4o, respectively.

## J  ANALYSIS OF OMNI-JUDGE

Omni-Judge is proposed as an economic but effective way to provide judgment for complicated mathematical problems accompanied by reference answers and tested solutions. In this part, we test Omni-Judge on the quality of generated judgment in two aspects: whether in a precise format and whether to generate correct judgment, represented by the rates of being successfully parsed (**Success**) and consistency rates with golden judgments (**Consistency**), as in Table 9. Note that judgments from GPT-4o are seen as the golden ones here. Furthermore, as discussed in Section 2.4, we experiment with various LLaMA-series models, and with upgraded versions, the comprehensive capabilities of models are generally acknowledged to improve. We find the performance of Omni-Judge also shows enhancements along this trend. In detail, we collect test data from several models tested in Table 3, 5 of which also provide data for training Omni-Judge (denoted as **Training with Homology Data**) and the rest 3 ones do not, while they demonstrate similar conclusions.

In terms of format accuracy, the **Success** rates are approximately 100% across different settings, while the **Consistency** rates are not. Omni-Judge model trained with LLaMA-3.1-8b-Instruct shows the best **Consistency** with GPT-4o in total, followed by the version of LLaMA-3-8b-Instruct. On the other hand, Omni-Judge exhibits the lowest scores based on LLaMA-2-7b-Chat. The high **Success** suggests that models tend to learn formats first, aligning with findings of format learning in Wang et al. (2022). However, to provide better judgments, the base models are required to effectively capture the optimization directions during fine-tuning or possess knowledge in this domain. Therefore, the highest **Consistency** of Omni-Judge on LLaMA-3.1-8b-Instruct is reasonable, and we consequently use it as the default version of Omni-Judge.

Another significant value of mathematical benchmarks is the provided ranking, which reflects the relative capability of each subject among all tested models. Therefore, it is meaningful to compare the ranking of model capabilities from Omni-Judge and GPT-4o, which suggests whether Omni-

Judge can be a practical judgment model offering reliable feedback. As in Figure 17, Omni-Judge provides outcomes similar to GPT-4o for each tested model, as well as an identical ranking of model capabilities according to the predicted accuracy, which proves its usability.

## K    THE CONSTRUCTION DETAILS OF OMNI-MATH-RULE

As mentioned in Section 2.4, we additionally filtered a subset named *Omni-MATH-Rule* for efficiently reliable rule-based evaluation.

We began by rewriting a new version of the rule-matching code using the Qwen2.5-math rule evaluation repository, primarily utilizing SymPy. Then we performed an initial filtering of the entire dataset of problems, which was done as follows:

We first applied the rule evaluation to the results from O1-mini, ensuring that all answers were within the "boxed" format using the system prompt. We then extracted the answers using the "last_boxed" method, followed by a rule evaluation, which was fully consistent with the MATH dataset.

We then compared the rule-based and GPT-4o-based results (given that GPT-4o-based evaluation can perform over 98% consistency with human evaluation). Based on their judgments of the policy model generation, we identified the following cases:

- Rule-based Correct and GPT-4o-based Correct: This indicates that the answer was accepted by both Rule-based and GPT-4o. We consider this case to have a high probability of evaluation correctness, and we labeled it as a "positive case."

- Rule-based False and GPT-4o-based False: This indicates that both Rule-based and GPT-4o rejected the answer. However, the rule-based rejection could be due to generalization issues, meaning it may not align with GPT-4o's rejection. We label this as an "uncertain case," and we need to use strict rules to filter out only those cases that can be reliably evaluated.

- Rule-based and GPT-4o-based judgments inconsistent: This case could suggest evaluation problems with either the GPT-4o evaluation or the rule evaluation. Therefore, we marked this as an "inconsistent case."

The statistics of the cases above can be shown as follows:

Table 10: statistics of Rule-based and GPT-4o-based Results

| Rule-based | GPT-4o-based | Count |
|---|---|---|
| Correct | Correct | 2053 |
| Correct | False | 66 |
| False | Correct | 545 |
| False | False | 1764 |

We found that less than 2% of cases (fewer than 66 out of 4428) were identified by GPT-4o as incorrect answers while still matching the rules correctly. We believe these instances reliably indicate evaluation errors by GPT-4o. Upon careful examination, we discovered that some cases experienced parsing failures, resulting in both the predicted and ground truth outputs being matched as empty strings, which accounted for 30 out of the 66 cases. After identifying this issue, we removed these cases from the testable dataset. Consequently, the actual number of instances categorized as "Rule-based: Correct and GPT-4o-based: False" is reduced to 36. Among these, several remain contentious; for example, GPT-4o indicated that a condition was omitted in the answer.

We then added the "positive cases" and "uncertain cases" to the testable set which indicates the cases are reliable with rule-based evaluation. The "inconsistent cases" were placed directly into the untestable set.

We applied strict rules (using regular expressions) to filter the testable set, removing cases that were more complex, such as open-ended text, multi-part LaTeX, function-based problems, or multiple

functions, retaining only simpler cases that the rules could match effectively, which can filter reliable cases in "uncertain case" as much as possible.

This filtering process resulted in 2925 problems suitable for rule-based evaluation. We then employed two PhD students to annotate the rule-based evaluated data. The task was to classify the type of answer based on the aforementioned experiments (number or tuples or latex and so on) and determine whether it could be evaluated by the rule system. We ensured that each problem was annotated by two students, and our cross-validation accuracy was 98% (2869/2925). Inconsistent cases were placed in the untestable set. This annotation process removed some tuples that were still unsuitable for evaluation and some complex LaTeX cases. Ultimately, We ended up with **a testable set of 2821 problems and an untestable set of 1607 problems**. We analyzed the difficulty distribution of the testable set, which is basically consistent with the distribution of our entire dataset.

To further validate the reliability of the testable set and the rule evaluation, we evaluated some models on the current subset. The rankings and the acc of the models were consistent with the evaluation results from GPT-4o.

Table 11: The leaderboard of total accuracy of Omni-MATH-Rule using rule-based evaluation and Omni-MATH using GPT-4o evaluation.

| Model | Acc @Rule 2821 | Acc @GPT-4o 4428 |
|---|---|---|
| o1-mini | 62.2% | 60.54% |
| o1-preview | 51.7% | 52.55% |
| qwen2.5-MATH-72b-Instruct | 35.7% | 36.20% |
| qwen2.5-MATH-7b-Instruct | 32.3% | 33.22% |
| GPT-4o | 29.2% | 30.49% |
| NuminaMATH-72b-cot | 27.1% | 28.45% |
| DeepseekMATH-7b-RL | 14.9% | 16.12% |

## L   DATA EXAMPLES OF OMNI-MATH

Domain: Mathematics -> Algebra -> Abstract Algebra -> Group Theory

Source: china_national_olympiad          Difficulty: 7

**Question**

Let $p$ be a prime. We arrange the numbers in ${\{1,2,\ldots ,p^2} \}$ as a $p \times p$ matrix $A = ( a_{ij} )$. Next we can select any row or column and add $1$ to every number in it, or subtract $1$ from every number in it. We call the arrangement [i]good[/i] if we can change every number of the matrix to $0$ in a finite number of such moves. How many good arrangements are there?

**Solution**

We call the arrangement "good" if we can change every number of the matrix to 0 in a finite number of such moves. We aim to determine the number of good arrangements. Assume we start with a good matrix and let $a_i$ and $b_j$ be the numbers of operations ("numbers add one" minus "numbers subtract one") on the $i$-th row and $j$-th column, respectively, to get a zero matrix.

Then we should have:
$$a_{ij} + a_i + b_j = 0.$$

Summing up all these equations, we obtain:
$$\sum_{i=1}^{p} a_i + \sum_{j=1}^{p} b_j = -p\left(\frac{p^2+1}{2}\right).$$

As a result, it implies the sum of numbers that lie on $A_{1\sigma(1)}, A_{2\sigma(2)}, \ldots, A_{p\sigma(p)}$ is constant for all $\sigma \in S_p$. By comparing these equations, we find that the first row of the matrix is just a translation of the second row, i.e., $A_{1i} - A_{2i}$ is constant for $1 \le i \le p$. This is true for any two other rows as well. Thus, the problem boils down to finding $a_1, a_2, \ldots, a_p \in \{1, 2, \ldots, p^2\}$ and $0 = x_1, x_2, \ldots, x_{p-1} \in \mathbb{Z}$ such that:

$$B_i = \{a_1 + x_i, a_2 + x_i, \ldots, a_p + x_i\}$$

represents the $i$-th row of matrix $M$ for $1 \le i \le p$, representing a partition of $\{1, 2, \ldots, p^2\}$. Without loss of generality, we can assume $1 = a_1 < a_2 < \ldots < a_p$ and $0 = x_1 < x_2 < \ldots < x_{p-1}$. We try to prove that either $x_i = i - 1$ (first column is $\{1, 2, \ldots, p\}$) or $a_i = i$ (first row is $\{1, 2, \ldots, p\}$). If $a_2 > 2$ and $x_2 > 1$, then neither the first row nor any other rows contain 2. We have two cases to consider: 1. $x_2 = 1$: This is straightforward. If $x_3 > 2$, then there is no possible position for 3, so $x_3 = 2$. Continuing in the same way, we find $x_i = i - 1$. 2. $a_2 = 2$: Let $k$ be the greatest number such that $a_i = i$ for all $1 \le i \le k$. Then where is $k + 1$? Yes, $x_2 = k$ and $k + 1, \ldots, 2k$ lie below $1, 2, \ldots, k$. Playing around with possible positions, we arrive at the matrix:

$$M = \begin{bmatrix} 1 & 2 & \ldots & k & 2k+1 & 2k+2 & \ldots & 3k & 4k+1 & \ldots \\ k+1 & k+2 & \ldots & 2k & 3k+1 & 3k+2 & \ldots & 4k & 5k+1 & \ldots \end{bmatrix}$$

This matrix implies $k \mid p$, which is a contradiction since $p$ is prime. Therefore, in both cases, we can suppose that $a_i = i$. This argument works symmetrically for columns as well. Hence, the number of good arrangements is:
$$2(p!)^2.$$

The answer is: $\boxed{2(p!)^2}$.

**Answer:**

The answer is: $\boxed{2(p!)^2}$.

Figure 18: An example from China National Mathematical Olympiad (CMO) in Omni-MATH.

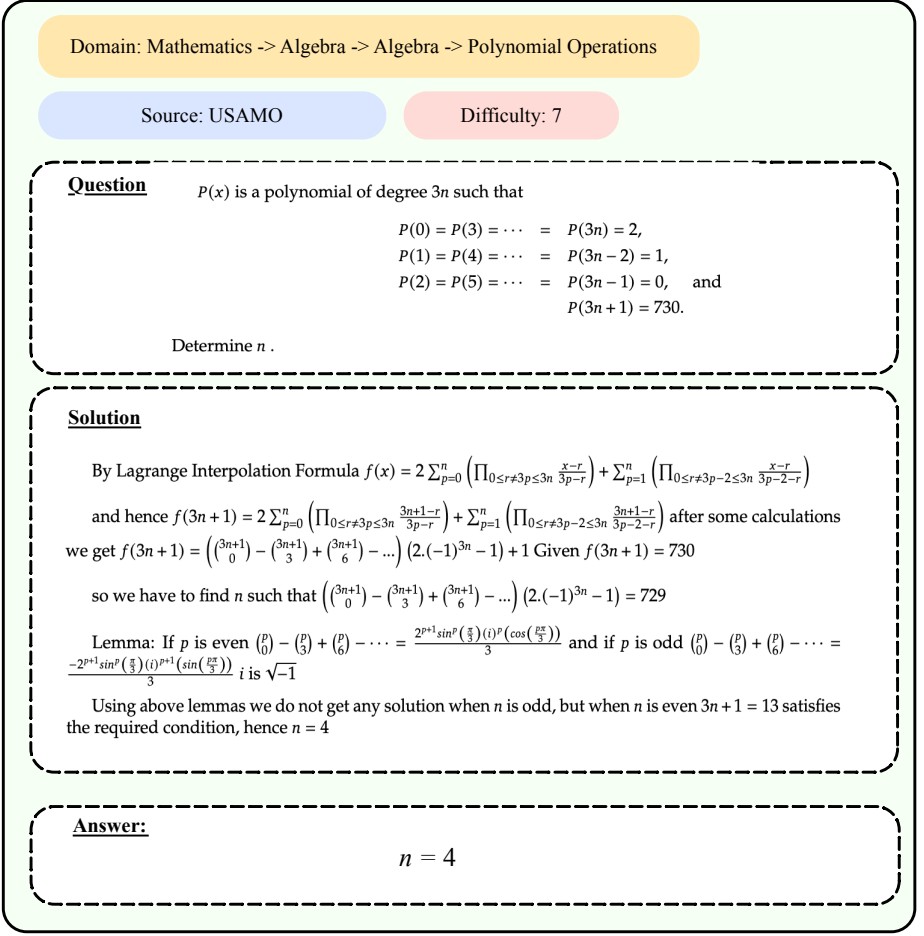

Figure 19: An example from USA Mathematical Olympiad (USAMO) in Omni-MATH.

