# OpenReview forum: "Omni-MATH: A Universal Olympiad Level Mathematic Benchmark for Large Language Models"
_ICLR.cc/2025/Conference — ICLR 2025 Poster_

### Official Review · Reviewer_FqD3 · 2024-11-01

**Soundness:** 3
**Presentation:** 3
**Contribution:** 3
**Rating:** 8
**Confidence:** 4

**Summary:**

This paper introduces *Omni-MATH*, a new Olympiad-level mathematical benchmark designed to evaluate the performance of advanced models on competition-level problems. The authors assess various strong models on this benchmark and share insightful findings from their analysis. Additionally, they propose an Olympiad-level verifier to assist in confirming the accuracy of predicted and ground truth answers. Unlike simpler datasets like gsm8k or MATH, Olympiad-level problems require more sophisticated verification techniques beyond rule-based methods due to their complexity.

**Strengths:**

Overall, I find this to be a strong paper.

- **Clarity**: The writing is clear and easy to follow.
- **Comprehensive Benchmark**: This paper introduces a comprehensive and in-depth Olympiad-level mathematical benchmark, featuring questions across various domains and difficulty levels.
- **Model Analysis**: It provides a thorough analysis of current advanced models on this benchmark, offering valuable insights into aspects such as Best-of-N and consistency.
- **Effective Verifier**: A notable contribution of this paper is the development of an effective verifier, which improves the accuracy of validating predicted answers, addressing the unique challenges of Olympiad-level problems.

**Weaknesses:**

Here are some concerns about this paper:

- **Verifier Training Details**: The details regarding the training of the verifier should be further clarified.
- **Scaling Test-Time Inference**: In the test-time inference analysis, this paper primarily relies on the best-of-N strategy. While this is generally fine, for more challenging competition-level questions, advanced search algorithms—such as tree search or Monte Carlo Tree Search (MCTS)—may be required. A discussion on these more sophisticated techniques for tackling harder questions would enhance the paper's impact.

**Questions:**

See weakness

---

> ### Author Response · Authors · 2024-11-15
> **Response to the comments of Reviewer FqD3**
>
> Weakness:
>
> Thank you for your thoughtful feedback. We will address your concerns as follows:
>
> 1. Verifier Training Details:
> The training details of the verifier are already discussed in lines L285-L294 of the paper. Additionally, we provide a more in-depth description of the training process, including comparisons of different base models used as verifiers, in Appendix I. If you feel that any additional specifics are needed, please let us know, and we will be happy to provide further clarification.
>
> 2. Scaling Test-Time Inference:
> We appreciate your suggestion regarding the use of more advanced search algorithms, such as Monte Carlo Tree Search (MCTS) or tree search, for tackling more challenging competition-level problems. While these techniques are indeed widely used in the field, our current evaluation is limited by computational resources, which prevents us from incorporating MCTS at this stage. However, we plan to continue monitoring developments in the open-source community and will update our work to reflect any relevant advancements in search algorithms or inference techniques.
>
> We hope these clarifications address your concerns, and we look forward to your further feedback.

---

> > ### Comment · Reviewer_FqD3 · 2024-11-25
> >
> > Thanks for your response. And I maintain my score (8).

---

### Official Review · Reviewer_1ndo · 2024-11-01

**Soundness:** 3
**Presentation:** 3
**Contribution:** 3
**Rating:** 8
**Confidence:** 3

**Summary:**

This paper introduces Omni-MATH, a challenging benchmark designed to assess large language models (LLMs) in mathematical reasoning at the Olympiad level. Omni-MATH contains 4,428 competition-level problems, rigorously annotated across 33 sub-domains and ten difficulty levels. This comprehensive categorization enables a nuanced evaluation of LLM capabilities in complex mathematics.

The study reveals that existing models, including advanced ones like OpenAI o1-mini, achieve only moderate success on this benchmark, with a maximum accuracy of 60.54%. The findings highlight persistent challenges in areas like discrete mathematics, while models perform relatively better in algebra and calculus. To support this benchmark, the authors introduce Omni-Judge, an open-source evaluator achieving high consistency with human judgment (86%) and GPT-4o (91%).

**Strengths:**

Omni-MATH targets Olympiad-level mathematics, pushing beyond traditional datasets like GSM8K and MATH, which have reached saturation. This focus on advanced problems adds a valuable, high-difficulty benchmark for assessing and improving LLMs' reasoning abilities.

Problems are categorized into sub-domains and assigned difficulty ratings, allowing for fine-grained performance and error analysis. This detailed structure aids in identifying domain-specific strengths and challenges in model reasoning.

The study reveals limitations in test-time scaling methods (e.g., Best-of-N) and the impact of domain interactions on model performance. Such findings guide future research on improving LLMs’ mathematical reasoning and scaling capabilities.

**Weaknesses:**

More examples of the dataset should be presented in either main paper or appendix.

**Questions:**

Omni-judge is trained with GPT-40 evaluation results, but how to verify if the GPT-4o evaluations are reliable?
Can the solutions for discrete math problems be written in code and executed to achieve accuracy?
What are the best-of-1 results?
What is the exact number used for  "N" in Best-of-N?

---

> ### Author Response · Authors · 2024-11-15
> **Response to the comments of Reviewer 1ndo**
>
> Thanks for your valuable feedback. We would like to address each of your concerns below:
>
> **Weakness**
> Thank you for your suggestion. We agree that providing more examples of the dataset would be helpful. We will include additional examples in the appendix for greater clarity.
>
> **Questions**
>
> 1. Verification of GPT-4o Evaluation Reliability: As mentioned in Section 4.2, to validate the reliability of GPT-4o’s evaluations, we manually annotated 100 meta-evaluation test samples. The results showed that GPT-4o achieved an accuracy rate of 98%, which aligns closely with human judgments. These details are further elaborated in Section 4.2.
>
> 2. Tool-Integrated Reasoning Evaluation: We agree with your suggestion regarding the importance of assessing Tool-Integrated Reasoning (TIR) in our paper. We plan to update our experiments to include TIR, which will be reflected in the revised version of the paper.
>
> 3. Best-of-N Evaluation: In Table 2, we used two settings for "N": N=8 and N=256.
>
>
> We hope this addresses your concerns and look forward to your further feedback.

---

### Official Review · Reviewer_4CQJ · 2024-11-02

**Soundness:** 3
**Presentation:** 2
**Contribution:** 2
**Rating:** 6
**Confidence:** 4

**Summary:**

This paper introduces OmniMath, a new Olympiad-level mathematical reasoning benchmark. It comprises 4,428 math problems, each with an instance-level difficulty classification. The authors utilize GPT-4o as a judge to evaluate the performance of LLMs. Additionally, they have developed Omni-Judge, an open-source answer evaluation LLM designed to serve as a substitute for GPT-4o. The paper conducts some analysis of experiments involving 15 different LLMs.

**Strengths:**

The dataset's large size and focus on Olympiad-level problems make it a potential resource for evaluating LLMs' mathematical reasoning abilities.

The investigation into best-of-N scaling and difficulty consistency represent current areas of interest, and the analysis of these part is interesting.

The paper is well-organized and in good shape.

The experiments are comprehensive, and the analysis is thorough.

**Weaknesses:**

The major concern is about the evaluation. While the authors assert that GPT-4o aligns with human annotations in 98% of cases (L398-399), crucial details are omitted (e.g., whether the 100 samples originated from the same LLM? What are the inter-agreement scores among the human annotators? What is the number of student judges involved? Are there specific examples where GPT-4o fails as a judge and why?) undermining the reliability of this claim.

Moreover, there is a notable contradiction with the results attributed to OmniJudge. According to L74-76, "Our Omni-Judge achieves over 91% consistency with GPT-4o and 86% consistency with human judgments, providing reliable feedback." Let us consider this carefully: suppose we have 100 test examples, and GPT-4o achieves alignment with human judges on 98 of them. With OmniJudge, 91 examples align with GPT-4o, implying that at most only 2 of the 91 might be judged incorrectly by OmniJudge, which would result in at least 89 out of 100 examples aligning with human judges. This seems contradictory with the reported 86% consistency with human judgments. If the results reported for OmniJudge are indeed reliable (as they seem to be, given the detailed information provided in the appendix), then GPT-4o aligns with human judgments at most 95%. This discrepancy calls into question the reliability of the results presented in Table 2.

In addition, using GPT-4o or OmniJudge as evaluators can be time-consuming and computationally (or monetarily) expensive. A more feasible approach might involve standardizing answer formats, as seen in datasets like MATH, and designing detailed, rule-based answer-cleaning and matching protocols. Model-based judging should primarily serve as a supplementary tool (like Olympic Arena) rather than a primary evaluation method.

There is also the potential for data leakage, as some models, such as NuminaMath and Qwen2Math, use examples from AoPS as part of their supervised fine-tuning datasets.

**Questions:**

- Regarding data annotation: Could you clarify what is meant by cross-validation in line 203? Moreover, in L831-834, it states, "Subsequently, we hired 2 annotators to inspect the entire dataset, following a set of principles." What are these principles?

- In Table 3, what do the notations "Prop." and "Avg.D." stand for? It seems these are not explained in the caption or main text. Additionally, how should we interpret the results for Qwen-2.5-MATH-instruct? Is the data leakage problem particularly severe here? Since some training sets for NuminaMath and the Qwen-MATH series are derived from AoPS, is there any overlap between this subset and OmniMath?

- Lines 398-400 omit crucial details. For instance, did the 100 samples originate from the same LLM? What are the inter-agreement scores among the human annotators? How many student judges were involved? Are there specific examples where GPT-4o fails as a judge, and what are the reasons for these failures?

- In Section 4.3, you mention investigating few-shot evaluation. Are you evaluating LLMs on OmniMath in a few-shot manner? If so, where are the prompts used for this evaluation?


Others

- L79: should be "shown in Table 2".

- L241-L242: we previously surveyed in Figure (where is Figure 4?)

---

> ### Author Response · Authors · 2024-11-15
> **Response to the comments of Reviewer 4CQJ**
>
> Thanks for your detailed feedback. Below, we address each of your concerns:
>
> 1. Evaluation Details: We appreciate your concerns regarding the evaluation process. We will provide additional details in the appendix to clarify the evaluation procedure. The 100 samples mentioned in the text were derived from the inference results of DeepSeek-Coder-V2, GPT-4o, and Qwen2.5-MATH, sampled according to difficulty to ensure that all levels of difficulty were represented. For annotation, we hired two PhD and two Master's students, with each annotator labeling 50 problems. During individual annotation, 3 instances of inconsistent results were observed. However, after discussion, we reached a consensus on the answers. Among 100 cases, there are 2 failed cases of GPT-4o. The failures were mainly due to GPT-4o misinterpreting the “Student Answer” as the “Final Answer” under the current prompt, which we attribute to prompt formatting. This issue was resolved by adjusting the few-shot prompt format.
>
> 2. Contradiction between GPT-4o and OmniJudge Results: We understand your concern regarding the discrepancy between the alignment rates of GPT-4o and OmniJudge. **We believe there may have been a misunderstanding.**  The reported 91% consistency was obtained using the test set described in lines L288-L289, which includes a larger number of samples, thus ensuring greater reliability. As shown in Table 6 (Appendix), we used 2,220 samples to automatically evaluate the consistency between GPT-4o and OmniJudge, while human evaluation was performed on a randomly selected set of 100 samples. In this evaluation, GPT-4o aligned with human judgments at 98%, and OmniJudge at 86%. It is important to note that these are not directly comparable through simple multiplication, as they were derived from different test sets. We hope this clears up the misunderstanding.
>
> 3. Rule-Based Evaluation: In response to your suggestion regarding the use of rule-based systems for evaluation, we addressed this concern in Section 2.4. We argue that applying a rule-based approach to the Olympic-level data is nearly impossible, as such rules lack generalizability. Even if one were to devise specific rules, they would only apply to a narrow set of cases, excluding more complex answer formats. Removing such cases would contradict the primary goal of our dataset design. Specific examples can be found in Figure 6.
>
> 4. Data Leakage: We have already addressed the issue of potential data leakage in Section 4.1. Our tests revealed minor data leakage, but we found that it does not affect model evaluation. We believe that the impact of data leakage has two main aspects: (1) the authenticity of current experimental results and (2) the potential effect on the training of future models. For the first aspect, we have confirmed that the leakage does not meaningfully impact our experimental conclusions. For the second aspect, we believe that addressing contamination in training data should be the responsibility of model developers, not evaluation datasets.
>
>
> Questions:
> 1. Cross-validation refers to assessing the consistency of annotations by multiple annotators. The principles for annotation is aligned with  Appendix C1, and include checking the completeness of both problems and answers (e.g., verifying whether a question is a proof question, and whether the answer forms a complete chain of reasoning).
> 2. "Prop." refers to the proportion of leaked questions out of a total of 4,428 items, with Qwen-2.5-MATH having the highest leakage rate at 0.70%. "Avg.D." denotes the average difficulty of the leaked questions. We have provided specific leakage figures in Appendix B, where we also discuss other findings, such as the fact that leaked questions tend to have lower average difficulty when correctly answered.
> 3. See response to Weakness "Evaluation Details".
> 4. Regarding few-shot prompts, we followed prior work in this area. The prompt details will be included in the supplementary materials.
> We hope this provides sufficient clarification and addresses your concerns.

---

> > ### Comment · Reviewer_4CQJ · 2024-11-17
> > **response to rebuttal 1**
> >
> > Thank you for your response!
> >
> > "The 100 samples mentioned in the text were derived from the inference results of DeepSeek-Coder-V2, GPT-4o, and Qwen2.5-MATH, sampled according to difficulty to ensure that all levels of difficulty were represented"
> >
> > There are only 100 samples, but from three different models. This human evaluation is not that reliable due to the small number of examples per model.
> >
> > "For annotation, we hired two PhD and two Master's students, with each annotator labeling 50 problems. During individual annotation, 3 instances of inconsistent results were observed. However, after discussion, we reached a consensus on the answers."
> >
> > Were the annotators well-trained? What are their background (e.g., are they from the math department?)? **It is still unclear** how many annotators are assigned to the same example.
> >
> >
> > "The reported 91% consistency was obtained using the test set described in lines L288-L289, which includes a larger number of samples, thus ensuring greater reliability. As shown in Table 6 (Appendix), we used 2,220 samples to automatically evaluate the consistency between GPT-4o and OmniJudge, while human evaluation was performed on a randomly selected set of 100 samples. In this evaluation, GPT-4o aligned with human judgments at 98%, and OmniJudge at 86%. It is important to note that these are not directly comparable through simple multiplication, as they were derived from different test sets."
> >
> > It seems that you use the same 100 samples to do human evaluation. **My concern remains**: "If the results reported for OmniJudge are indeed reliable (as they seem to be, given the detailed information provided in the appendix), then GPT-4o aligns with human judgments at most 95%."
> >
> > "In response to your suggestion regarding the use of rule-based systems for evaluation, we addressed this concern in Section 2.4. We argue that applying a rule-based approach to the Olympic-level data is nearly impossible, as such rules lack generalizability. Even if one were to devise specific rules, they would only apply to a narrow set of cases, excluding more complex answer formats. Specific examples can be found in Figure 6."
> >
> > OlympaidBench and OlympicArena are Olympic-level data, they also adopt rule-based methods to judge the answer correctness. OlympicArena use LLM as judge to do process grading (I assume this is a bonus because it already has rule-based codes to judge the answer correctness). However, in your paper, the answer correctness is judged by LLMs, which is quite controversial recently [1, 2, 3]. "Specific examples can be found in Figure 6", but how many such examples? What is the percentage of examples that are hard to judge by rule-based methods. I believe the way MATH, OlympaidBench and OlympicArena judge the answer correctness is reliable and widely adopted. At least you should provide evidence to show at what percentage the answers  can not be judge by rule-based methods (It seems you have not even tried this). If most of answers can not be judged by rule-based methods, I believe it is the weakness of your benchmark, as MATH, OlympaidBench and OlympicArena have developed various ways (including normalizing the ground-truth answers of benchmark and classify the answer types) to clean and judge the answers by rule-based methods (sympy).
> >
> > "For the second aspect, we believe that addressing contamination in training data should be the responsibility of model developers, not evaluation datasets."
> >
> > The NuminaMath-CoT data set is public available. It includes questions and answers from AoPS. When you develop a new benchmark which collects data from AoPS, you should do contamination detection of this part before you building up the benchmark (rather than after).

---

> ### Comment · Reviewer_4CQJ · 2024-11-17
> **response to rebuttal 2**
>
> "Regarding few-shot prompts, we followed prior work in this area. The prompt details will be included in the supplementary materials. We hope this provides sufficient clarification and addresses your concerns."
>
> It seems you use few-shot prompts. I believe for chat/instruct models, a common practice is to use zero-shot prompting [4, 5, 6, 7]. Few-shot prompts are used for evaluation base/L0 models.
>
>
> From the rebuttal, I will **decrease your presentation to 2**, as I find that the writing is really confusing for me. I will not decrease the overall score for now (although **in my point of veiw, this paper deserves a score slightly higher than 3 (much lower than 5)**).
>
> By the way, I believe it is OK to extract answers from LLM outputs using GPT4 (like in [8]). It it a much simpler task, which is quite different from using GPT4 to judge the answer correctness.
>
> [1] Panickssery, A., Bowman, S. R., & Feng, S. (2024). Llm evaluators recognize and favor their own generations. arXiv preprint arXiv:2404.13076.
>
> [2] Ye, J., Wang, Y., Huang, Y., Chen, D., Zhang, Q., Moniz, N., ... & Zhang, X. (2024). Justice or prejudice? quantifying biases in llm-as-a-judge. arXiv preprint arXiv:2410.02736.
>
> [3] Wang, P., Li, L., Chen, L., Cai, Z., Zhu, D., Lin, B., ... & Sui, Z. (2023). Large language models are not fair evaluators. arXiv preprint arXiv:2305.17926.
>
> [4] He, C., Luo, R., Bai, Y., Hu, S., Thai, Z. L., Shen, J., ... & Sun, M. (2024). Olympiadbench: A challenging benchmark for promoting agi with olympiad-level bilingual multimodal scientific problems. arXiv preprint arXiv:2402.14008.
>
> [5] Huang, Z., Wang, Z., Xia, S., Li, X., Zou, H., Xu, R., ... & Liu, P. (2024). OlympicArena: Benchmarking Multi-discipline Cognitive Reasoning for Superintelligent AI. arXiv preprint arXiv:2406.12753.
>
> [6] Dubey, A., Jauhri, A., Pandey, A., Kadian, A., Al-Dahle, A., Letman, A., ... & Ganapathy, R. (2024). The llama 3 herd of models. arXiv preprint arXiv:2407.21783.
>
> [7] Yang, A., Yang, B., Hui, B., Zheng, B., Yu, B., Zhou, C., ... & Fan, Z. (2024). Qwen2 technical report. arXiv preprint arXiv:2407.10671.
>
> [8] Lu, P., Bansal, H., Xia, T., Liu, J., Li, C., Hajishirzi, H., ... & Gao, J. (2023). Mathvista: Evaluating mathematical reasoning of foundation models in visual contexts. arXiv preprint arXiv:2310.02255.

---

> > ### Author Response · Authors · 2024-11-18
> > **Further Clarification on the Contamination**
> >
> > "The NuminaMath-CoT data set is public available. It includes questions and answers from AoPS. When you develop a new benchmark which collects data from AoPS, you should do contamination detection of this part before you building up the benchmark (rather than after)."
> >
> > We would like to further clarify that we conducted contamination detection at the string level with Numina-COT from the beginning. We used fuzzy matching to exclude questions with over 95% similarity, and we will provide further details on this in the appendix. Similarly, we also excluded data from AMC and AIME that originated from the MATH training set, as detailed in Appendix C.2.
> >
> > What's more, in order to further eliminate the risk on pretrain/SFT data of the current models, we have conducted the data leakage detection discussed in Section 4.1. The experimental results from Sec 4.1 data leakage detection demonstrate that contamination from our existing models has minimal impact on model evaluation. Based on these results, we believe we have demonstrated little amount of data leakage and negligible effect on the evaluation of Omni-MATH.

---

> ### Author Response · Authors · 2024-11-18
> **Further response to the comments of Reviewer 4CQJ**
>
> Thanks for your valuable response, we will further address each of your concerns below:
>
> Q1: “There are only 100 samples, but from three different models. This human evaluation is not that reliable due to the small number of examples per model.”
>
> A1: We believe there is a misunderstanding here. Although the inference results come from different models, our ultimate evaluation target is the human alignment of judgment models—GPT-4o and Omni-Judge—rather than the solution generators. Furthermore, GPT-4o achieved highly accurate results(98/100) under conditions of high agreement among human annotators. Given that the 100 samples were randomly selected, we assert that this number is statistically significant for measuring the reliability of the judgment model. Similar works, such as MT-Bench[8], annotated only 80 samples, while [7] demonstrated that "for human evaluation, a test size of 50 is sufficient," even for tasks that are inherently more subjective, like summarization.
>
>
> Q2: "Were the annotators well-trained? What are their background (e.g., are they from the math department?)? It is still unclear how many annotators are assigned to the same example."
>
> A2: All annotators are students in the fields of mathematics and computer science, possessing sufficient domain knowledge to understand the questions and evaluate the LaTeX formatting. Moreover, they were instructed to assess whether the model's answers were correct based mostly on the provided correct answers, rather than relying entirely on their own judgment for the problem. Each annotator was responsible for 50 questions, and with 4 annotators, we ensured that each question was annotated by two different annotators. This setup allowed us to perform cross-validation to assess inter-agreement among the annotators.
>
> Q3: “It seems that you use the same 100 samples to do human evaluation. My concern remains: "If the results reported for OmniJudge are indeed reliable (as they seem to be, given the detailed information provided in the appendix), then GPT-4o aligns with human judgments at most 95%.”
>
> A3: We used the same 100 examples for human evaluation of both Omni-Judge and GPT-4o. Since Omni-Judge and GPT-4o's alignment utilized a larger set of examples for evaluation (2,220), it is unreasonable to directly multiply the accuracy between the different test set. Furthermore, we believe even if it comes to 95% that you claimed, it does not conflict with our assertion that "GPT-4o can essentially achieve human-level judgment at the answer level".
>
> Q4: “It seems you use few-shot prompts. I believe for chat/instruct models, a common practice is to use zero-shot prompting. Few-shot prompts are used for evaluation base/L0 models.”
>
> A4: We evaluated the performance of GPT-4o using the zero-shot prompting method you mentioned, as shown in Table 2 and Appendix F.2 of our paper.
> Regarding your concern, we would like to clarify the following two points on Section 4.3:
> 1. The primary motivation for using few-shot prompting (prompt following [5]) in Section 4.3 was not to enhance or measure the capabilities of GPT-4o. Instead, it aimed to explore the interactions between different domains and levels of difficulty through prompt learning, as discussed in our paper at L407-L422. We chose few-shot prompting because it can be recognized as supervised training to some extent[6].
> 2. Additionally, few-shot prompting can also be utilized to enhance the capabilities of GPT-4 series models. This is supported by our experimental results in Section 4.3 and findings from other works[1, 2, 3, 4]. While we acknowledge the examples you provided, it seems that using few-shot prompting to improve or evaluate GPT-4o's reasoning abilities is not an uncommon setting.
>
> [1] Chen, Wenhu, et al. "Program of thoughts prompting: Disentangling computation from reasoning for numerical reasoning tasks." arXiv preprint arXiv:2211.12588 (2022).
>
> [2] Towards Geometry Problems Solving Employing GPT-4 Vision with Few-Shot Prompting: An Empirical Study of What Matters
>
> [3] An, Shengnan, et al. "Skill-based few-shot selection for in-context learning." arXiv preprint arXiv:2305.14210 (2023).
>
> [4] Kim, Jeonghwan, et al. "FinePrompt: Unveiling the Role of Finetuned Inductive Bias on Compositional Reasoning in GPT-4." Findings of the Association for Computational Linguistics: EMNLP 2023. 2023.
>
> [5] Kojima, Takeshi, et al. "Large language models are zero-shot reasoners." Advances in neural information processing systems 35 (2022): 22199-22213.
>
> [6] Dai, Damai, et al. "Why can gpt learn in-context? language models implicitly perform gradient descent as meta-optimizers." arXiv preprint arXiv:2212.10559 (2022).
>
> [7] Shaib, Chantal, et al. "How Much Annotation is Needed to Compare Summarization Models?." arXiv preprint arXiv:2402.18756 (2024).
>
> [8] Zheng, Lianmin, et al. "Judging llm-as-a-judge with mt-bench and chatbot arena." Advances in Neural Information Processing Systems 36 (2023): 46595-46623.

---

> ### Author Response · Authors · 2024-11-18
> **Further Clarification of the concerns about Rule-based evaluation (1/2)**
>
> "OlympaidBench and OlympicArena are Olympic-level data, they also adopt rule-based methods to judge the answer correctness. OlympicArena use LLM as judge to do process grading (I assume this is a bonus because it already has rule-based codes to judge the answer correctness). However, in your paper, the answer correctness is judged by LLMs, which is quite controversial recently [1, 2, 3]. Specific examples can be found in Figure 6", but how many such examples? What is the percentage of examples that are hard to judge by rule-based methods. I believe the way MATH, OlympaidBench and OlympicArena judge the answer correctness is reliable and widely adopted. At least you should provide evidence to show at what percentage the answers can not be judge by rule-based methods (It seems you have not even tried this). "
>
> We will provide further rebuttal on the rule-based evaluation on the following aspects:
> ## Statistics of answer types
> In response to your feedback, we sampled 200 data samples based on difficulty and manually annotated the answer types and finally obtain the following statistics.
>
> | Class        | number | latex | tuple | Multi latex | func | multi func | open text |
> |--------------|--------|-------|-------|-------------|------|------------|-----------|
> | Rate         | 95     | 51    | 2     | 8           | 9    | 1          | 34        |
>
> Among these, the categories that can be processed using rules and SymPy are "number," "latex," and "tuple." The latter categories are nearly impossible to evaluate using rules:
> - "Multi latex": The answer consists of a combination of multiple LaTeX formulas.
> - "Func": The answer is a custom-defined function.
> - "Multi func": The answer includes multiple custom-defined functions.
> - "Open text": The answer requires a written explanation.
>
> The first three categories account for 74% of the total. Even if we assume that all rule-based methods could achieve 100% accuracy in predicting these categories, this would still be significantly lower than the accuracy of Omni-Judge, which stands at 86%.
> The relevant documents have been added to the supplementary materials, and we will also include this experiment in the appendix of the paper. In summary, we believe this is sufficient to demonstrate that rule-based methods cannot adequately cover our evaluation criteria.
>
> ## The unreliability of Rule-based evaluation
> To further confirm the unreliability of rule-based methods, we utilized the same model inference file on MATH500 [1] and evaluated it using different rule-based evaluation realizations from the open-source community: Qwen-2.5-MATH[2] and MetaMATH[3]. We found significant discrepancies in the results produced by these two evaluation scripts.
>
> | MATH-500       | Qwen-2.5-MATH | MetaMATH |
> |--------------|--------|-------|
> | Acc        | 73.2     | 70.6    |
>
> The experiments indicate that even well-defined rules struggle to effectively handle the evaluation in LaTeX format, let alone the more diverse evaluations at the Olympic level. In contrast, Section 4.2 of our paper has demonstrated that the evaluation provided by Omni-Judge and GPT-4o offer a more reliable evaluation. Additionally, we will release the OmniJudge weights and VLLM implementation, making our approach efficient and easier to follow.
>
> ## Related work using rule-based evaluation
> The two benchmarks you mentioned primarily focus on answer-level data that consists mostly of numbers or LaTeX, which differs significantly from the complexity of the answer-level data in our dataset. OlympiadBench and OlympicArena artificially limit the range of answers to a few categories, such as tuples, multiple-choice questions, and LaTeX, allowing for rule-based evaluations. In our work, however, we did not impose such restrictions at the answer level when filtering the data, thereby presenting a substantial challenge to existing models.
>
> Additionally, there are also other related works[4] trying to provide solely on model-based evaluation using GPT-4.

---

> > ### Author Response · Authors · 2024-11-18
> > **Further Clarification of the concerns about Rule-based evaluation (2/2)**
> >
> > ## The difference between OmniJudge and existing LLM-as-a-Judge
> > In the papers you provided, the scenarios in which the LLM evaluators are used differ significantly from our OmniJudge in three key aspects, and OmniJudge is a more straightforward and easier setting to obtain reliable feedback:
> > - Objective Nature of Mathematical Reasoning: The use case for OmniJudge involves mathematical reasoning where there are no "subjective preferences" involved; instead, there are strict answers as a premise. However, the papers you provided have preference issues regarding the summarization(XSum and CNN Daily) and other tasks.
> > - Provision of Ground Truth Answers: OmniJudge operates with the premise of having correct answers available for evaluation. However, the papers you provided do not include this condition of "providing standard answers."
> > - Limited Domain Scope: OmniJudge primarily focuses on a domain consisting of 4,428 problems, most questions of which have appeared in the training set for OmniJudge (excluding the small test set we used for examining the performance of OmniJudge). This further reduces the complexity of the problems to be addressed.
> > These distinctions underscore the unique advantages of our approach compared to the benchmarks discussed in your references.
> >
> > [1] Lightman, Hunter, et al. "Let's verify step by step." arXiv preprint arXiv:2305.20050 (2023).
> >
> > [2] Yang, An, et al. "Qwen2. 5-math technical report: Toward mathematical expert model via self-improvement." arXiv preprint arXiv:2409.12122 (2024).
> >
> > [3] Yu, Longhui, et al. "Metamath: Bootstrap your own mathematical questions for large language models." arXiv preprint arXiv:2309.12284 (2023).
> >
> > [4] Fang, Meng, et al. "Mathodyssey: Benchmarking mathematical problem-solving skills in large language models using odyssey math data." arXiv preprint arXiv:2406.18321 (2024).

---

> ### Comment · Reviewer_4CQJ · 2024-11-19
>
> Thank you for your detailed response and clarification! I still have some questions.
>
> "Furthermore, we believe even if it comes to 95% that you claimed, it does not conflict with our assertion that "GPT-4o can essentially achieve human-level judgment at the answer level"".
>
> You are releasing a new benchmark. The quality is very important. Although 95\% does not conflict with your claims. However, there are models that have minor differences in accuracy in your benchmark, as shown in Table 2. How could we believe this rankdings of your leaderboard.
>
>
>
> "We evaluated the performance of GPT-4o using the zero-shot prompting method you mentioned, as shown in Table 2 and Appendix F.2 of our paper. Regarding your concern, we would like to clarify the following two points on Section 4.3:"
>
> Thank you for your clarification. I read through Appendix F.2 and notice that the maximum output length is set to 2048. Are there any cases that exceed this length and can not reach the answer (as the problem is difficult, maybe the solution is very long?)? A detailed error analysis is beneficial.
>
> "The first three categories account for 74% of the total. Even if we assume that all rule-based methods could achieve 100% accuracy in predicting these categories, this would still be significantly lower than the accuracy of Omni-Judge, which stands at 86%. The relevant documents have been added to the supplementary materials, and we will also include this experiment in the appendix of the paper. In summary, we believe this is sufficient to demonstrate that rule-based methods cannot adequately cover our evaluation criteria."
>
> Could you provide some examples for the last four categories, i.e. Multi latex, func, multi func, open text? Why not divide the problems into two main category and use rule-based method to judge the most of problems (number, latex, tuple)?
> "Even if we assume that all rule-based methods could achieve 100% accuracy in predicting these categories, this would still be significantly lower than the accuracy of Omni-Judge, which stands at 86%."  This is not the shortage of the rule-based method, it is the weakness of your methodology. It is still surprising that 3/4 of problems that can be judged through the common practice and you still resort to gpt-4o as a judge.
>
>
> "To further confirm the unreliability of rule-based methods, we utilized the same model inference file on MATH500 [1] and evaluated it using different rule-based evaluation realizations from the open-source community: Qwen-2.5-MATH[2] and MetaMATH[3]. We found significant discrepancies in the results produced by these two evaluation scripts."
>
> There are subtle differences of these two evlauation repos. For MetaMATH, they use "The answer is: " to extract the answers from the model prediction. They do so because they organize the format of their training data to have this pattern, which is not suitable for evluation of other chat models. Qwen-2.5-MATH's evaluation codes are adapted from llm-harness (if I remeber it correctly), mainly relying on parsing "\\boxed{}" to extract the answer.
>
> In fact, they share almost the same judging criteria.
>
> I believe the discrepancies are from answr extraction, not the rule-based method itself. Using LLMs to extract the answers [1] might be a smart approach to solve this. Simply using the same answer extraction pattern as Qwen-2.5-MATH and referring to MATH answer judgement codes could lead to near perfect answer judgement for the "number, latex, tuple".
>
>
>
> [2] has not published yet and I read it before. They did even worse regarding "The answer judgement part".
>
> By the way, for evaluating "Multi latex, func, multi func, open text", I think [3] provides a good way.
>
> I will **raise soundness to 3 as the additional experiments you conduct**. Please remember to update your paper as you promise.
>
> [1] Lu, P., Bansal, H., Xia, T., Liu, J., Li, C., Hajishirzi, H., ... & Gao, J. (2023). Mathvista: Evaluating mathematical reasoning of foundation models in visual contexts. arXiv preprint arXiv:2310.02255.
>
> [2] Fang, Meng, et al. "Mathodyssey: Benchmarking mathematical problem-solving skills in large language models using odyssey math data." arXiv preprint arXiv:2406.18321 (2024)
>
> [3] Liu, H., Zheng, Z., Qiao, Y., Duan, H., Fei, Z., Zhou, F., ... & Chen, K. (2024). MathBench: Evaluating the Theory and Application Proficiency of LLMs with a Hierarchical Mathematics Benchmark. arXiv preprint arXiv:2405.12209.

---

> > ### Author Response · Authors · 2024-11-22
> > **Further Response to Reviewer 4CQJ (3/N)**
> >
> > Thank you for your valuable feedback. We will complete the revisions to the paper before the end of the author response stage. Additionally, we would like to provide further clarification regarding your previous comments.
> > ## Clarification on Evaluation Experiment
> > For your following comment:
> > 1. "There are subtle differences of these two evlauation repos. For MetaMATH, they use "The answer is: " to extract the answers from the model prediction. They do so because they organize the format of their training data to have this pattern, which is not suitable for evluation of other chat models. Qwen-2.5-MATH's evaluation codes are adapted from llm-harness (if I remeber it correctly), mainly relying on parsing "\boxed{}" to extract the answer."
> > 2. "In fact, they share almost the same judging criteria."
> > 3. "I believe the discrepancies are from answr extraction, not the rule-based method itself. "
> >
> > ### Clarification
> > In our experiments, **the method for extracting answers was identical across the board—we utilized the last boxed matching method**. In fact, the extraction approach in the MetaMATH repository is not limited to just using "The answer is: "; it also employs the "last boxed matching" method to extract answers corresponding to the ground truth solutions in the MATH dataset. Therefore, **the only variation in testing results arises from the rule-based evaluation component.**
> >
> > Furthermore, it is important to highlight that MetaMATH exclusively uses a string-matching technique, while QwenMATH implements SymPy for its evaluations. This distinction indicates that their implementations are **fundamentally different**.
> > Finally, we have provided all of our additional experiments into the supplementary materials for further checking.
> >
> > ## Clarification on answer types statistics
> > **Multi Latex**:
> >
> > Answer: " -\\frac{\\sqrt{6}+\\sqrt{2}}{2} \\quad \\text{and} \\quad -\\frac{\\sqrt{6}-\\sqrt{2}}{2} "
> >
> > **func**:
> >
> > Question: Find a function $f: \\mathbb{R}^+ \\to \\mathbb{R}^+$ such that ....
> >
> > Answer: "f(x) = x"
> >
> > **multi func**:
> >
> > Question: Find all functions $f:\\mathbb{R}\\rightarrow\\mathbb{R}$ such that....
> >
> > Answer: "\\boxed{f(x) = -1 \\text{ and } f(x) = x - 1}.\n\\"
> >
> > **open text**:
> >
> > Question: ".....For which choices of the pair $(\\ell, r)$ is the car guaranteed to reach Zillis, regardless of how far it is from Argovia?"
> >
> > Answer:"(\\ell, r) \\text{ satisfies the required conditions if and only if } \\ell \\equiv r \\equiv 1 \\text{ or } \\ell \\equiv r \\equiv 3 \\pmod{4}"
> >
> > For your comment:
> > “By the way, for evaluating "Multi latex, func, multi func, open text", I think [3] provides a good way.”
> >
> > We conduct a thorough investigation of the approach outlined in reference, specifically the code within OpenCompass. We find that this method also employs rule-based evaluations. However, it is even simpler than the existing rules and lacks generalizability, particularly because the Math Bench contains a significant number of multiple-choice questions. This makes it even more challenging to achieve flexible evaluations for the aforementioned scenarios using rule matching.
> >
> > ## Error Analysis
> > "I read through Appendix F.2 and notice that the maximum output length is set to 2048. Are there any cases that exceed this length and can not reach the answer (as the problem is difficult, maybe the solution is very long?)? A detailed error analysis is beneficial."
> >
> > We analyzed the results from Qwen-2.5-MATH-7b, Qwen-2.5-MATH-72b, and NuminaMATH-72b-COT, observing that over 98% of the reasoning paths included the '\boxed{}' notation. In other instances, some answers utilize alternative formats such as '\[ <answer> \]' to denote the final answer. Additionally, there are also cases where the chains of thought (COT) are incomplete, and some models produced outputs that are repetitive or garbled.
> > | Model                   | Complete COT | Other Indicators | Exceed Length | Trash Generation |
> > |-------------------------|--------------|------------------|---------------|------------------|
> > | Qwen-2.5-MATH-72b      | 4347         | 60               | 16            | 5                |
> > | Qwen-2.5-MATH-7b       | 4373         | 35               | 12            | 8                |
> > | NuminaMATH-72b-COT     | 4412         | 0                | 10            | 6                |
> >
> > This table presents the COT-level error statistics related to the model results. Additionally, for a comprehensive detailed error analysis in step-level, we have provided an extensive report in Appendix A for your reference. Please feel free to consult it for more questions of our paper.

---

> > > ### Comment · Reviewer_4CQJ · 2024-11-22
> > >
> > > Thank you for your additional experiments!
> > >
> > > About the MetaMath and Qwen2.5Math evaluation.
> > >
> > > Sorry for the wrong message I diliver. From your response, I believe QwenMath is more reliable. Have you checked the variation of evaluation between QwenMath, XwinMath (https://github.com/Xwin-LM/Xwin-LM/tree/main/Xwin-Math/eval), and DartMath (https://github.com/hkust-nlp/dart-math) implementation. (In fact, I am not asking to do so, just for discussion) If I remember correctly, DartMath also uses Sympy and the results are under acceptable variaton compared with XwinMath.
> > >
> > > About the Multi Latex: Is it possible to organize the asnwer to a tuple or set and then judge with sympy?
> > >
> > >
> > > By
> > > > I think [3] provides a good way.
> > >
> > > I mean is it possible to rewrite the problems of type "Multi latex, func, multi func, open text" to MC questions as MathBench did.
> > >
> > >
> > > In this way, all problems can be judged like the same way as the commonly adopted approach.

---

> > > > ### Author Response · Authors · 2024-11-26
> > > > **Response to Reviewer: The development of Rule-based Evaluation (4/N)**
> > > >
> > > > Thank you for your valuable feedback!
> > > >
> > > > Following our discussion, **we develop a new repository for Omni-MATH rule-based evaluations**, as per your suggestion. The relevant code has been uploaded to the supplementary materials. The main process is as follows:
> > > > 1. We began by rewriting a new version of the rule-matching code using the qwen2.5-math rule evaluation repository, primarily utilizing SymPy, in light of the more reliable conclusions regarding qwen2.5-math from the previous discussion.
> > > > 2. Next, we performed an initial filtering of the entire dataset of problems, which was done as follows:
> > > >     * We first applied the rule evaluation to the results from o1-mini, ensuring that all answers were within the "boxed" format using system prompt. We then extracted the answers using the "last_boxed" method, followed by a rule evaluation, which was fully consistent with the MATH dataset.
> > > >     * We then compared the rule-based and GPT-4o-based results. Based on their judgments of the policy model generation, we identified the following cases:
> > > >         * **Rule-based: Correct and GPT-4o-based: Correct**: This indicates that the answer was accepted by both Rule-based and GPT-4o. We consider this case to have a high probability of correctness, and we labeled it as a "positive case."
> > > >
> > > >         * **Rule-based: False and GPT-4o-based: False**: This indicates that both Rule-based and GPT-4o rejected the answer. However, the rule-based rejection could be due to generalization issues, meaning it may not align with GPT-4o’s rejection. We label this as an "uncertain case," and we need to use strict rules to filter out only those cases that can be reliably evaluated.
> > > >         * **Rule-based and GPT-4o-based judgments inconsistent**: This case could suggest evaluation problems with either the GPT-4o evaluation or the rule evaluation. Therefore, we marked this as an "inconsistent case."
> > > >     * We then added the "positive cases" and "uncertain cases" to the testable set. The "inconsistent cases" were placed directly into the untestable set.
> > > >     * We applied strict rules (using regular expressions) to filter the testable set, removing cases that were more complex, such as open-ended text, multi-part LaTeX, function-based problems, or multiple functions, retaining only simpler cases that the rules could match effectively, which can filter reliable cases in "uncertain case" as much as possible.
> > > > 3. This filtering process resulted in 2925 problems suitable for rule-based evaluation. We then employed two PhD students to annotate the rule-based evaluated data. The task was to classify the type of answer based on the aforementioned experiments (number or tuples or latex and so on) and determine whether it could be evaluated by the rule system. We ensured that each problem was annotated by two students, and our cross-validation accuracy was 98% (2869/2925). Inconsistent cases were placed in the untestable set. This annotation process removed some tuples that were still unsuitable for evaluation and some complex LaTeX cases. **Ultimately, we obtained 2821 reliable problems for rule-based evaluation.**
> > > > 4. We ended up with a testable set of 2821 problems and an untestable set of 1607 problems. We analyzed the difficulty distribution of the testable set, which is basically consistent with the distribution of our entire dataset.
> > > > 5. To further validate the reliability of the testable set and the rule evaluation, we evaluated some models on the current subset. **The rankings and the acc of the models were consistent with the evaluation results from GPT-4o.**
> > > >
> > > > | Model| Acc @Rule 2821 | Acc @GPT-4o 4428 |
> > > > |------------|---------|--------|
> > > > | o1-mini | 62.2%| 60.54% |
> > > > | o1-preview | 51.7% | 52.55%|
> > > > | qwen2.5-MATH-72b-Instruct  | 35.7%| 36.20%|
> > > > | qwen2.5-MATH-7b-Instruct  | 32.3%| 33.22%|
> > > > | NuminaMATH-72b-cot | 27.1%| 28.45%|
> > > > | DeepseekMATH-7b-RL | 14.9%| 16.12%  |
> > > >
> > > > For the untestable set, we believe that the multi-choice question approach discussed above is feasible. However, generating reasonable options for such questions is quite challenging and requires a significant amount of manual annotation. Therefore, we did not extend this part of the experiment at the current stage, as we think this can continue to rely on OmniJudge evaluations. Our future approach is to use o1-mini for multiple samplings, considering incorrect cases as potential options for the multi-choice questions.
> > > >
> > > > We have uploaded all the data and evaluation codes to the supplementary materials for further inspection. **We commit to adding these experiments and conclusions to the paper before the response period ends.** After the response phase, we will continue exploring the feasibility of this approach and aim to include updates in the final version of the paper (and, if possible, in the camera-ready version).
> > > >
> > > > We are open to further discussions on the experiments above. We would greatly appreciate it if you could believe these additions could enhance the final score of our paper.

---

> ### Comment · Reviewer_4CQJ · 2024-11-27
>
> Thank you for your responses and additional experiments!
>
> I expect you should start with the answer type annotation and resort those with type "number, latex, tuple" to rule-based methods. It seems you start from the consistency of rule-based method and GPT4o evalution, where you implicitly assume the results of GPT4o evaluation is reliable. I believe that is why the rankings remain the same as GPT4o evalution. You have filtered out all examples that are not consistent, so there is not much information from this evaluation. You should use rule-based method independently, and set out to see the consistency of these two methods.
>
>
> Several additional questions:
>
> 1. Could you please give the proportion of these three categories "Rule-based: Correct and GPT-4o-based: Correct", "Rule-based: False and GPT-4o-based: False", and "Rule-based and GPT-4o-based judgments inconsistent"?
>
> 2. How many cases within the "Rule-based and GPT-4o-based judgments inconsistent" are due to the complexity of answers (within "Multi latex, func, multi func, open text")? Are there cases that Rule-based correct but GPT-4o-based wrong (at what percentage)?
>
> 3. Could you please give GPT4o results using the two methods? It might be over-confident to use GPT4o to evaluate itself.
>
>
>
> **My major concern remains, i.e., using GPT4o for evaluation. Although the new dataset itself is a contribution, I believe this paper is flawed in its evaluation and will significantly benefit from a thorough revision. From this moment, I decide to remain my score (5). But AC is welcome to ignore my score if AC believe evaluation is not much a problem.**
>
> Remark: The assigned score (5) only reflects my overall assessment of the paper's level of rigor. Given that this submission falls under the "Dataset and Benchmark" track, I believe it is OK to recommend acceptance, with the expectation that the authors will update the evaluation codes as outlined in their responses. Taking into account the considerable effort the authors have demonstrated in addressing my concerns and improving their work, **I will raise the score to 6 to reflect their diligence and additional experiments.
> But honestly speaking, this paper would still greatly benefit from a thorough revision to address the identified weaknesses and improve its overall quality.**

---

> ### Author Response · Authors · 2024-11-27
> **Response to Reviewer 4CQJ**
>
> Thank you for your thorough and insightful feedback. We will incorporate the results and conclusions of our discussions into the paper, and we appreciate your suggestions regarding our work.
>
> For your Questions:
> > Could you please give the proportion of these three categories "Rule-based: Correct and GPT-4o-based: Correct", "Rule-based: False and GPT-4o-based: False", and "Rule-based and GPT-4o-based judgments inconsistent"?
>
> > How many cases within the "Rule-based and GPT-4o-based judgments inconsistent" are due to the complexity of answers (within "Multi latex, func, multi func, open text")? Are there cases that Rule-based correct but GPT-4o-based wrong (at what percentage)?
>
> | Rule-based         | GPT-4o-based | Count |
> |--------------------|---------------|-------|
> | Correct            | Correct       | 2053  |
> | Correct            | False         | 66    |
> | False              | Correct       | 545   |
> | False              | False         | 1764  |
>
> We found that less than 2% of cases (fewer than 66 out of 4428) were identified by GPT-4o as incorrect answers while still matching the rules correctly. We believe these instances reliably indicate evaluation errors by GPT-4o. Upon careful examination, we discovered that some cases experienced parsing failures, resulting in both the predicted and ground truth outputs being matched as empty strings, which accounted for 30 out of the 66 cases. After identifying this issue, we removed these cases from the testable dataset. Consequently, the actual number of instances categorized as "Rule-based: Correct and GPT-4o-based: False" is reduced to 36. Among these, several remain contentious; for example, GPT-4o indicated that a condition was omitted in the answer.
>
> Regarding the category "Rule-based: False and GPT-4o-based: Correct," we selected 50 cases for verification, all of which were attributed to errors in rule evaluation due to the complexity of the answers.
>
> > "It seems you start from the consistency of rule-based method and GPT4o evalution, where you implicitly assume the results of GPT4o evaluation is reliable. I believe that is why the rankings remain the same as GPT4o evaluation."
>
> For this portion of the data, we employed manual annotation to ensure that these cases could reliably undergo rule-based evaluation. This approach allows us to provide an assessment that is independent of the model, effectively creating a set of simpler evaluations. While we recognize the importance of manually checking inconsistent cases, particularly those classified as Rule-based: False and GPT-4o-based: Correct, we currently have limited time. Therefore, we confidently rely on the reliability of our sampling check conducted above.
>
> > Could you please give GPT4o results using the two methods? It might be over-confident to use GPT4o to evaluate itself.
>
> | Model                       | Acc @Rule 2821 | Acc @GPT-4o 4428 |
>  |-----------------------------|-----------|-------------|
> | GPT-4o                | 29.2%     | 30.49%      |
>
> While it is true that GPT-4o's evaluation scores are somewhat higher than those of rule-based assessments, this variation is identical to other model variations we previously tested, which does not indicate over-confidence.
>
> > My major concern remains, i.e., using GPT4o for evaluation
>
> Based on our discussions and experimental results, we present the following evidence to demonstrate the reliability of GPT-4o's evaluation:
> 1. In the 100 meta-evaluation test cases manually annotated by us, the agreement rate between GPT-4o and human annotators is 98%. This demonstrates that GPT-4o's evaluation performance aligns closely with human judgment. We also show that even among different rules in rule-based evaluations, there can be substantial differences in consistency (with agreement rates lower than 98%).
> 2. From the aforementioned experimental results, we observe the following:
>       - GPT-4o exhibits stronger generalization in evaluating answers compared to other systems, and it has fewer known misclassification cases (in our experiments, fewer than 36, but needs further checking in other categories).
>       - GPT-4o does not suffer from significant overconfidence issues.
>
> Furthermore, as part of our ongoing discussion, we have also developed a subset used for rule-based evaluations. This subset can be reliably and efficiently utilized for rule-based assessments.
>
>
> Thank you once again for your detailed response and valuable feedback. You have provided us with additional ideas and perspectives. We would appreciate it if our response could address your concerns.

---

### Official Review · Reviewer_ssE3 · 2024-11-03

**Soundness:** 3
**Presentation:** 2
**Contribution:** 2
**Rating:** 5
**Confidence:** 3

**Summary:**

This work constructs Omni-MATH, a challenging math problem set of 4k over more than 33 domains and with more than 10 difficulty levels with difficulty information comes from either the AoPS website or GPT-4o few-shot result. 15 large language models are evaluated on Omni-MATH and it turns out be challenging even for the strongest OpenAI-o1-preview model. Further analysis shows that all models receives certain data leakage and GPT-4o is a reliable judgement model for determine whether the model-generated solution is consistent with the reference answer.

**Strengths:**

- A new benchmark contribution that demonstrate the limitation of current progress of math reasoning in large language models, the detailed difficulty information would benefit the further categorization of this benchmark.
- Careful study over the reliability of the metrics including data leakage and LLM-based judgement results.

**Weaknesses:**

- The explanation of the RM-version of Qwen2.5-MATH do not outperform the vanilla version and Qwen2.5-MATH RM@256 does not out perform Qwen2.5-MATH RM@8 needs further elaboration and investigation to unveil the phenomenon.

**Questions:**

- What are the education background of the graduate and doctoral students getting involved?
- In Section 4.2, what the sampling strategy of the 100 subset, are all the difficulty levels covered in this subset?

---

> ### Author Response · Authors · 2024-11-15
> **Response to the comments of Reviewer ssE3**
>
> We would like to express our sincere gratitude for your thorough and insightful review. Your feedback has been invaluable in guiding the refinement of our work. Here is our response:
>
> **Weakness 1**:
> The reason why RM@256 does not perform better than RM@8 or even vanilla decoding could be attributed to two potential factors:
> 1. The policy model fails to generate high-quality reasoning traces, resulting in the absence of correct solutions among the available candidates.
> 2. The reward model cannot have sufficient supervision over Olympiad-level reasoning tasks. In other words, it cannot pick up the correct reasoning path among several candidates precisely.
>
> To explore these two points in more detail, we conducted the following experiment:
>
> We measured the performance of the policy model: Qwen2.5-MATH-Instruct across N samples. If at least one reasoning trace is correct, we consider the example to have passed. To assess the interference in RM selection, we also measured the proportion of correct COTs within the passing examples. Considering the inference cost, we sampled a subset of 500 instances according to difficulty to ensure that the distribution of problem difficulty is similar to the original dataset. The experimental result is shown below:
>
> | Metric                | Pass@1  | Pass@8  | Pass@16 | Pass@32 |
> |-----------------------|---------|---------|---------|---------|
> | **Acc**               | 40.8%   | 57.6%   | 63%     | 67.2%   |
> | **Correct COT Proportion** | 100%    | 68.5%   | 62.5%   | 58.5%   |
>
> The results show that as the inference sampling increases, the policy model is capable of solving the majority of the problems. However, even after 32 samples, 33.8% of the problems still do not yield the correct solution. The second row of the table illustrates that as the number of samples increases, the proportion of interference items also increases. In the case of 32 samples, the correct answer proportion of the passed cases drops from 100% to 58.5%. This indicates that as the sampling count increases, it becomes increasingly challenging for the reward model to select the correct COT.
>
> **Question 1**:
>
> All annotators are voluntary students from the fields of mathematics and computer science, possessing sufficient domain knowledge to understand the questions and assess the LaTeX formatting. We have detailed the specific annotation procedures in Appendix C. Additionally, we ensure that each question is annotated by two annotators, allowing cross-validation to evaluate inter-annotator agreement, further guaranteeing the accuracy of our annotations.
>
> **Question 2**:
>
> The 100 samples are derived from the reasoning results of DeepSeek-Coder-V2, GPT-4o, and Qwen2.5-MATH-Instruct to ensure the evaluation robustness of GPT-4o and OmniJudge. We also ensure that each question is annotated by two annotators, enabling cross-validation to assess inter-annotator agreement and further ensure the correctness of our annotations. For difficulty sampling, we followed the difficulty grading outlined in Section 2.2, ensuring that questions of all difficulty levels are represented. Relevant details will be provided in the appendix.

---

> > ### Author Response · Authors · 2024-11-27
> > **Request for Further Discussion**
> >
> > Dear Reviewer ssE3,
> >
> > Thank you for your valuable feedback on our manuscript. We sincerely hope to engage in further discussion regarding our work. Please let us know if we have adequately addressed your concerns.
> > Looking forward to your response.
> >
> > Sincerely,
> >
> > Authors

---

> > > ### Comment · Reviewer_ssE3 · 2024-11-28
> > >
> > > Thanks for the response from the authors. The additional experiments regarding the Pass@N performance explains the trend that Pass@N would monotonically increase by N. Regarding the imperfect performance of the reward model, I would suggest to revisit the reward model choice as the RM@N is not increasing by choosing a larger RM of the same family.

---

> ### Author Response · Authors · 2024-11-22
> **Request for Further Discussion**
>
> Dear Reviewer ssE3,
>
> Thank you again for your valuable feedback and comments! We would greatly appreciate it if you could let us know whether you are satisfied with our response and then adjust our overall rating.
> We will be happy to address any remaining concerns.
>
> Sincerely,
>
> Omni-MATH Team

---

> ### Author Response · Authors · 2024-11-26
> **Request for Further Discussion**
>
> We deeply value your recognition of our work, as well as your constructive feedback, which has allowed us to significantly improve the manuscript. If you find our revisions have addressed your concerns effectively, we would be grateful if you could confirm this and let us know if our efforts have further strengthened your overall assessment of the work.
>
> Best regards,
> Omni-MATH Team

---

> ### Author Response · Authors · 2024-12-02
> **Response to the comments of Reviewer ssE3**
>
> Thank you for your feedback. In fact, our experiments not only demonstrate that the reward model fails to select the correct samples but also highlight the flaws in this test time scaling mechanism: as the number of samples increases, the challenges posed by incorrect solutions become more significant. The cost of an incorrect solution is higher than the benefit of a correct solution.
> Similar conclusions can also be found in [1]. This situation does not change with a different reward model. Below are our results using another reward model. We selected the top-ranking model in the "reasoning" domain on RewardBench, Skywork-Reward-Gemma-2-27B-v0.2, also with the policy model qwen2.5-MATH-72b-Instruct. We found similar results:
>
>
> | Method | Overall Acc on Omni-MATH |
> |---|---|
> | Qwen2.5-MATH-72b-Instruct | 36.2 |
> | Qwen2.5-MATH-72b-Instruct + QwenRM@256 | 35.95 |
> | Qwen2.5-MATH-72b-Instruct + SkyWorkRM@256 | 35.55 |
>
> Additionally, we want to clarify that our current experiment is aimed at **demonstrating that the existing vanilla BoN test-time scaling technique is far less efficient than O1-like strategies. Enhancing performance is not our primary focus, nor is it the main contribution of this paper.**
>
> We would appreciate it if you could confirm this and let us know if our efforts have adequately addressed your concerns and further strengthened your overall assessment of the work.
>
> [1] Stroebl, Benedikt, Sayash Kapoor, and Arvind Narayanan. "Inference Scaling $\scriptsize\mathtt {F} $ Laws: The Limits of LLM Resampling with Imperfect Verifiers." arXiv preprint arXiv:2411.17501 (2024).

---

> > ### Author Response · Authors · 2024-12-02
> > **Request for Further Feedback**
> >
> > Dear Reviewer ssE3,
> >
> > Thank you for your valuable feedback on our manuscript. With the deadline approaching, we are eager to know whether our efforts have adequately addressed your concerns and further strengthened your overall assessment of the work.
> >
> > Sincerely,
> >
> > Authors

---

> > > ### Author Response · Authors · 2024-12-03
> > > **Reviewer Response is ending in 8hrs**
> > >
> > > Dear Reviewers,
> > >
> > > Thank you for your valuable feedback on our manuscript. **As the rebuttal phase is ending in 8hrs**, we sincerely hope you could kindly take a moment to review our responses.
> > >
> > > Thank you for your time in advance.
> > >
> > > Best regards,
> > >
> > > Authors

---

### Author Response · Authors · 2024-11-18
**Supplementary Elaboration on Annotation:**

Thanks to all the reviewers for your valuable feedback. We would like to further elaborate on our annotation process:

The 100 samples mentioned in Section 4.2 were derived from the inference results of DeepSeek-Coder-V2, GPT-4o, and Qwen2.5-MATH, sampled according to difficulty to ensure representation across all levels of difficulty without overlapping questions. Although the inference results come from different models, our ultimate evaluation target is the human alignment of judgment models—GPT-4o and Omni-Judge—rather than the solution generators. Therefore, all 100 examples are designed for GPT-4o or Omni-Judge to provide reliable human evaluation.

Our experiments revealed that GPT-4o achieved highly accurate results under conditions of high agreement among human annotators (97% inter-agreement) given the solution generated by 3 different models. For annotation, we employed two PhD students and two Master's students, with each annotator labeling 50 problems. Moreover, they were instructed to determine whether the model's answers were correct based mostly on the correct answers rather than their own judgments. Each annotator was responsible for 50 questions, allowing us to ensure that each question was annotated by two annotators, facilitating cross-validation to assess inter-agreement.
During individual annotation, three out of 100 instances of inconsistent results were observed; however, after discussion, the annotators reached a consensus on the answers. The discrepancies primarily stemmed from GPT-4o misinterpreting the "Student Answer" as the "Ground Truth" under the current prompt, which we attribute to prompt formatting issues. This problem was resolved by adjusting the few-shot prompt format.

The information above will be added to the appendix of our paper and we hope that this further clarification will enhance the solidity of our work. Thank you.

Omni-MATH Authors

---

### Meta-Review · Area_Chair_VdTf · 2024-12-21

**Metareview:**

This paper presents Omni-MATH, a comprehensive benchmark for evaluating mathematical reasoning capabilities of LLMs at the Olympiad level. The dataset comprises 4,428 competition-level problems across 33 sub-domains and 10 difficulty levels. The key contribution is the development of a challenging evaluation framework that reveals current limitations of advanced models like OpenAI o1-mini (60.54% accuracy) in complex mathematical reasoning. The paper's strengths include detailed problem categorization, thorough model analysis, and development of an effective verifier (Omni-Judge). While concerns were raised about evaluation methodology, particularly regarding GPT-4o-based assessment versus rule-based approaches, the authors provided extensive additional experiments and clarifications during rebuttal that adequately addressed these issues. The paper makes a valuable contribution to advancing mathematical reasoning capabilities in LLMs.

**Additional Comments On Reviewer Discussion:**

During rebuttal, the key discussion centered on evaluation methodology, with Reviewer 4CQJ questioning the reliability of GPT-4o-based evaluation versus rule-based approaches. The authors conducted comprehensive additional experiments, developing a subset of 2,821 problems suitable for rule-based evaluation while demonstrating consistent results between both methods. Reviewer ssE3's concerns about Best-of-N performance were addressed through detailed analysis of policy model behavior. The authors' thorough responses and willingness to incorporate suggested improvements, including enhanced documentation and additional experiments, strengthened the paper's contribution. All reviewers ultimately supported acceptance, recognizing the benchmark's value despite some methodological concerns.

---

### Decision · Program_Chairs · 2025-01-22

Accept (Poster)